# Structural insights into respiratory complex I deficiency and assembly from the mitochondrial disease-related *ndufs4*⁻/⁻ mouse

Zhan Yin [1,2], Ahmed-Noor A Agip [1,3], Hannah R Bridges [1,4 ✉] & Judy Hirst [1 ✉]

## Abstract

**Respiratory complex I (NADH:ubiquinone oxidoreductase) is essential for cellular energy production and NAD⁺ homeostasis. Complex I mutations cause neuromuscular, mitochondrial diseases, such as Leigh Syndrome, but their molecular-level consequences remain poorly understood. Here, we use a popular complex I-linked mitochondrial disease model, the *ndufs4*⁻/⁻ mouse, to define the structural, biochemical, and functional consequences of the absence of subunit NDUFS4. Cryo-EM analyses of the complex I from *ndufs4*⁻/⁻ mouse hearts revealed a loose association of the NADH-dehydrogenase module, and discrete classes containing either assembly factor NDUFAF2 or subunit NDUFS6. Subunit NDUFA12, which replaces its paralogue NDUFAF2 in mature complex I, is absent from all classes, compounding the deletion of NDUFS4 and preventing maturation of an NDUFS4-free enzyme. We propose that NDUFAF2 recruits the NADH-dehydrogenase module during assembly of the complex. Taken together, the findings provide new molecular-level understanding of the *ndufs4*⁻/⁻ mouse model and complex I-linked mitochondrial disease.**

**Keywords** Complex I; Cryo-EM; Leigh Syndrome; Mitochondria; NADH:Ubiquinone Oxidoreductase
**Subject Categories** Molecular Biology of Disease; Structural Biology

## Introduction

Mammalian complex I (NADH:ubiquinone oxidoreductase) is central to cellular metabolism, with key roles in NAD⁺ homeostasis, respiration, and oxidative phosphorylation. It couples NADH oxidation in the mitochondrial matrix to ubiquinone reduction in the inner membrane, reoxidizing the NAD⁺ pool to sustain metabolism, and captures the energy from the redox reaction to transfer protons across the inner membrane and create the proton-motive force that drives ATP synthesis. Together with

the ability of complex I to catalyze reactive oxygen species production, these key metabolic roles make complex I dysfunctions some of the most frequent causes of primary mitochondrial diseases, including Leber's hereditary optic neuropathy (LHON) and Leigh syndrome (Koene et al, 2012; Fiedorczuk and Sazanov, 2018; Padavannil et al, 2022; Fassone and Rahman, 2012).

Complex I is a 'boot-shaped' enzyme, with a hydrophilic domain in the mitochondrial matrix that contains the cofactors for redox catalysis, and a membrane-bound domain that transports the protons. Mammalian complex I comprises 45 subunits: 14 conserved core subunits that are essential for catalysis (seven in the hydrophilic domain and seven in the membrane domain) and 31 supernumerary subunits, including two copies of NDUFAB1 (Brandt, 2006; Zhu et al, 2016; Padavannil et al, 2022; Hirst et al, 2003). Single-particle electron cryo-microscopy (cryo-EM) has revolutionized our structural knowledge of complex I, and high-resolution structures have now been determined from several mammalian species [*Bos taurus* (cow, bovine complex I), *Ovis aries* (sheep, ovine complex I), *Mus musculus* (mouse, murine complex I) and *Sus scrofa* (pig, porcine complex I)] (Chung et al, 2022b; Gu et al, 2022; Kravchuk et al, 2022; Bridges et al, 2023) as well as from the complex I-model eukaryote, the yeast *Yarrowia lipolytica* (Parey et al, 2019). NADH is oxidized by a flavin mononucleotide in subunit NDUFV1 at the top of the hydrophilic domain, within a subdomain called the N-module (subunits NDUFV1, NUDUFV2, NDUFS1, NDUFA2, and NDUFV3). Then, electrons are transferred along a chain of seven iron-sulfur (FeS) clusters to bound ubiquinone, which enters 'head first' from the membrane and transits along a long, narrow channel to approach the final cluster in the chain (cluster N2). The mechanism by which ubiquinone reduction then drives proton transfer across the membrane at distant sites remains controversial (Kampjut and Sazanov, 2022; Chung et al, 2022a).

NDUFS4 (also known as 18 kDa, ND-18, AQDQ, or NUYM) is a supernumerary subunit in the hydrophilic domain of complex I, located where the N-module is bound to the rest of the complex (Fig. 1A,B). In the mammalian enzyme, it comprises N- and C-terminal coils flanking a central 4-strand β-sheet and α-helix. The N-terminus reaches down to NDUFA6 on the lower section of the hydrophilic domain (also known as the Q-module), while the

¹The Medical Research Council Mitochondrial Biology Unit, University of Cambridge, Keith Peters Building, Cambridge Biomedical Campus, Cambridge, UK. ²Present address: Department of Biochemistry, University of Cambridge, Tennis Court Road, Cambridge CB2 1GA, UK. ³Present address: Max-Planck-Institute of Biophysics, Frankfurt 60438, Germany. ⁴Present address: Structura Biotechnology Inc., Toronto, Canada. ✉E-mail: hbridges@structura.bio; jh480@cam.ac.uk

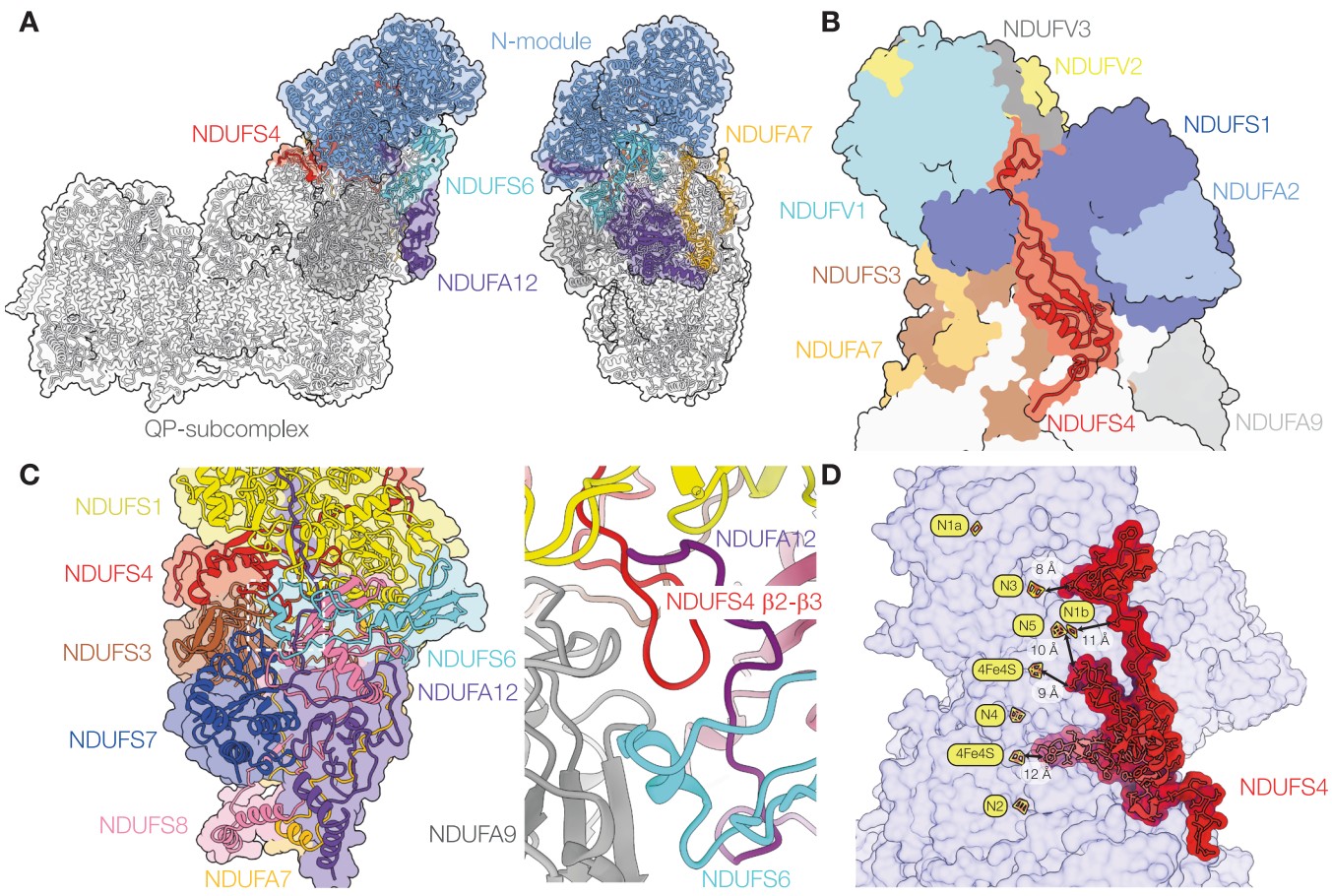

**Figure 1. The NDUFS4 subunit in wild-type complex I from mouse-heart mitochondria.**

(A) Overview structures of complex I viewed from the membrane and from the heel showing the locations of NDUFS4 (red), the N-module (blue) and subunits NDUFS6 (cyan), NDUFA12 (purple) and NDUFA7 (gold). (B) View of the front of the hydrophilic domain showing NDUFS4 (red) and the surrounding subunits. (C) Subunits surrounding NDUFA12 on the heel of the complex, and structural elements interacting with the β2-β3 loop of NDUFS4 that passes underneath NDUFS1 to form a junction of NDUSF4, NDUFA12, NDUFA6 and NDUF9 (right, close-up of the boxed region from a different viewpoint to focus on the loop). (D) NDUFS4 approaches four of the seven FeS clusters in the FeS chain between the FMN and the ubiquinone-binding site. Figure created using PDB: 6ZR2.

C-terminus runs up over the N-module, and the central β-sheet and α-helix domain is sandwiched between the N-module (NDUFS1) and the Q-module (predominantly NDUFS3) (Fig. 1B), anchoring them together. In *Escherichia coli* complex I, which lacks NDUFS4, the region occupied by the central domain is occupied instead by a C-terminal extension of NDUFS1 (Padavannil et al, 2022; Kolata and Efremov, 2021). These elements of NDUFS4 are all on the front of the hydrophilic domain (we use the boot analogy to identify different parts of the enzyme), while the β2-β3 loop runs through the domain, under NDUFS1, to form interactions with subunits located on the heel, including NDUFS6 and NDUFA12 (Fig. 1C). Thus, with structural elements interacting with both the N-module and the Q-module, a central domain in the 'crevice' between them, and a loop inserted through the enzyme to act like a rivet, NDUFS4 is key for complex I stability. In this location, NDUFS4 is within 12 Å of five iron-sulfur clusters (Fig. 1D). Finally, mammalian NDUFS4 contains a canonical protein kinase A site at Ser131, near its C-terminus, which has been proposed to be important for both catalysis and homeostasis (Signorile et al, 2002; Rasmo et al, 2010), including resistance of the N-module to degradation in mouse-brain astroglial cells (Jimenez-Blasco et al, 2020). However,

phosphorylation of this buried residue has not yet been observed in any structure.

Mammalian complex I is assembled from a set of assembly intermediates, comprising different segments of the enzyme, that are built stepwise from subunits and assembly factors then joined together to form the mature complex as the assembly factors are released (Guerrero-Castillo et al, 2017; Stroud et al, 2016). In the model of Nijtmans and coworkers (Guerrero-Castillo et al, 2017), the N-module intermediate (core subunits NDUFV1, NDUFV2, NDUFS1 and supernumerary subunit NDUFA2) is added to the nascent complex in the final stage of assembly, along with four supernumerary subunits (NDUFS4, NDUFS6, NDUFA12 and NDUFA7) that are the focus of the work presented here, and supernumerary subunits NDUFV3 (N-module), NDUFA6 and NDUFAB1 (Q-module), and NDUFA11 (proximal membrane domain). Subunits NDUFS4, NDUFS6, NDUFA12 all form contacts between the N-module and the Q-module, and are all conserved in complex I from the α-proteobacterium *Paracoccus denitrificans* (Yip et al, 2011), an ancient relative of the mitochondrion. As the N-module is added to the nascent complex, assembly factor NDUFAF2 (also known as B17.2 L) is released and

replaced by its paralogous subunit NDUFA12 (also known as B17.2) (Lazarou et al, 2007).

Many clinical studies have identified *ndufs4* mutations in severe pediatric neurological disorders such as Leigh or Leigh-like Syndrome, with symptoms including hypotonia, abnormal ocular movement and visual impairment, psychomotor arrest or regression and respiratory failure (Budde et al, 2003; Ortigoza-Escobar et al, 2016; van de Wal et al, 2022; Koene et al, 2012). Analyses of patient fibroblasts and muscle biopsies uncovered substantial loss of complex I activity, along with characteristic features in blue-native PAGE (BN-PAGE) analyses: lower abundance of intact complex I plus appearance of an ~830 kDa subcomplex that lacks in-gel NADH-oxidation activity, indicating the absence of the N-module (Scacco et al, 2003; Assouline et al, 2012; Ugalde, 2004). Underlining their functional relationship, mutations in *ndufs6*, *ndufa12*, and *ndufaf2* have all also been detected in cases of Leigh Syndrome (Spiegel et al, 2009; Ostergaard et al, 2011; Hoefs et al, 2009; Kahlhöfer et al, 2021).

The *ndufs4* gene is highly conserved between humans and mice, and most known pathogenic mutations lead to the absence of the whole NDUFS4 protein (van de Wal et al, 2022), so *ndufs4*⁻/⁻ mice have proved a powerful and representative model for studying NDUFS4-linked complex I deficiency, mechanisms of pathogenicity, and intervention testing (Kruse et al, 2008; Breuer et al, 2013; van de Wal et al, 2022). Both whole-body and neuron-specific *ndufs4*⁻/⁻ mice develop early-onset Leigh-like phenotypes (van de Wal et al, 2022), and complex I activity and stability decrease substantially in all the tissues studied in whole-body *ndufs4*⁻/⁻ mice (Calvaruso et al, 2012). Intriguingly, *ndufs4*⁻/⁻ mice exposed to prolonged hypoxia show marked improvements in their condition, and extended lifespan compared to mice kept under normoxia (Jain et al, 2016). Previous detailed proteomic studies in *ndufs4*⁻/⁻ mice observed the expected ~830 kDa subcomplex in BN-PAGE analyses, substantially lower complex I subunit levels in brain than in liver, heart, kidney, and skeletal muscle, loss of more than 90% of subunit NDUFA12, and increased complex I assembly factors, most notably NDUFAF2. Matching results were obtained from *ndufs4*⁻/⁻ mouse embryonic fibroblasts and human patient cells, and it was proposed that absence of NDUFS4 induces absence of NDUFA12, and that NDUFAF2 stabilizes the *ndufs4*⁻/⁻ enzyme (Adjobo-Hermans et al, 2020).

Here, we use single-particle cryo-EM to determine the structure of complex I from *ndufs4*⁻/⁻ mouse-heart mitochondria, building on our previous structures of the wild-type enzyme (Agip et al, 2018; Bridges et al, 2020) and the ND6-P25L variant (Yin et al, 2021). The ND6-P25L mouse is also a complex I-linked mitochondrial disease variant (though with milder phenotype than *ndufs4*⁻/⁻) that acts through a subtle 'deactivating' change in the membrane domain to produce an enzyme incapable of reverse electron transfer. In the absence of substrates, mammalian complex I converts gradually from an 'active' ready-to-go resting state to the 'deactive' pronounced resting state, which requires substrates to reactivate it (Kotlyar and Vinogradov, 1990; Chung et al, 2022a). Deactivation also occurs in the wild-type mouse enzyme, but much more slowly, whereas the *ndufs4*⁻/⁻ complex is observed completely in the active state. It is also incompletely assembled, its N-module is not stably attached, and its composition is heterogenous. By considering the structural determinants of catalytic activity and stability in *ndufs4*⁻/⁻ mouse complex I we advance understanding

of its molecular phenotype and the basis of complex I-linked mitochondrial dysfunctions.

## Results

### Deletion of NDUFS4 decreases complex I stability and its activity in membranes

To assess the structural integrity of complex I without the NDUFS4 subunit, we first investigated the complexes present in mitochondrial membranes from *ndufs4*⁻/⁻ (Kruse et al, 2008) and wild-type mouse heart mitochondria by BN-PAGE (Fig. 2A). The gels were probed with nitroblue tetrazolium (NBT), which reacts at the NADH-reduced complex I flavin site: *ndufs4*⁻/⁻ membranes exhibited a markedly less intense NADH:NBT-active complex I band at ~1 MDa than wild-type membranes, plus a new NADH:NBT-active band at much lower mass not evident in wild-type samples. Unusually strong Coomassie-staining was also observed at the position corresponding to the migration of complex V (ATP synthase).

BN-PAGE analyses of *ndufs4*⁻/⁻ and wild-type mouse heart mitochondrial membranes were then subjected to complexomic analyses to characterize the subunit compositions of complex I-related species, and to confirm the identity of *ndufs4*⁻/⁻-specific bands (Figs. 2B and EV1). For the wild-type membranes, 41 of the 44 complex I proteins were detected in the ~1 MDa complex I band. Only subunits ND3, ND4L, and ND6, which are hydrophobic proteins with few tryptic peptides, were not detected. For the *ndufs4*⁻/⁻ membranes, two new *ndufs4*⁻/⁻-specific bands were observed at ~800 kDa and ~200 kDa (by including species with known masses as standards in the BN-PAGE analyses their masses were estimated as 812 and 199 kDa, respectively, Fig. EV1). Mapping the 32 complex I subunits detected in the ~800 kDa band to the structure of the wild-type enzyme showed that they represent a substantial subcomplex containing the whole of the membrane domain and the lower (Q-module) section of the hydrophilic domain, but lacking the top section of the hydrophilic domain that contains the active site for NADH oxidation (the NADH-dehydrogenase module or N-module) (Figs. 2B and EV1). The N-module subunits (NDUFV1, NDUFV2, NDUFV3, NDUFS1, and NDUFA2) were identified in the NADH:NBT-active ~200 kDa band. The ~800 kDa subcomplex contains the subunits that catalyze ubiquinone reduction and proton pumping, so we refer to it as the QP subcomplex. An additional three subunits, NDUFS6, NDUFA12, and NDUFA7, usually located at the interface between the N-module and QP subcomplex, were detected elsewhere in the analyses, not associated with either. Note that the complexome profiles for each protein are normalised to the intensity in the highest abundance slice in each lane so they do not reflect the absolute abundance, only the relative abundance within the lane. Two complex I assembly factors, ACAD9 and NDUFAF2, also comigrate with the QP subcomplex, although NDUFAF2 is present in a similar position in the wild-type membrane analyses, one gel slice below the intact complex. The complexome analyses thus indicate that the absence of NDUFS4 weakens or prevents the attachment of the N-module (and adjacent subunits) to the complex. We note that ACAD9, in its active FAD-bound form, plays an important role in β-oxidation of fatty acids in the

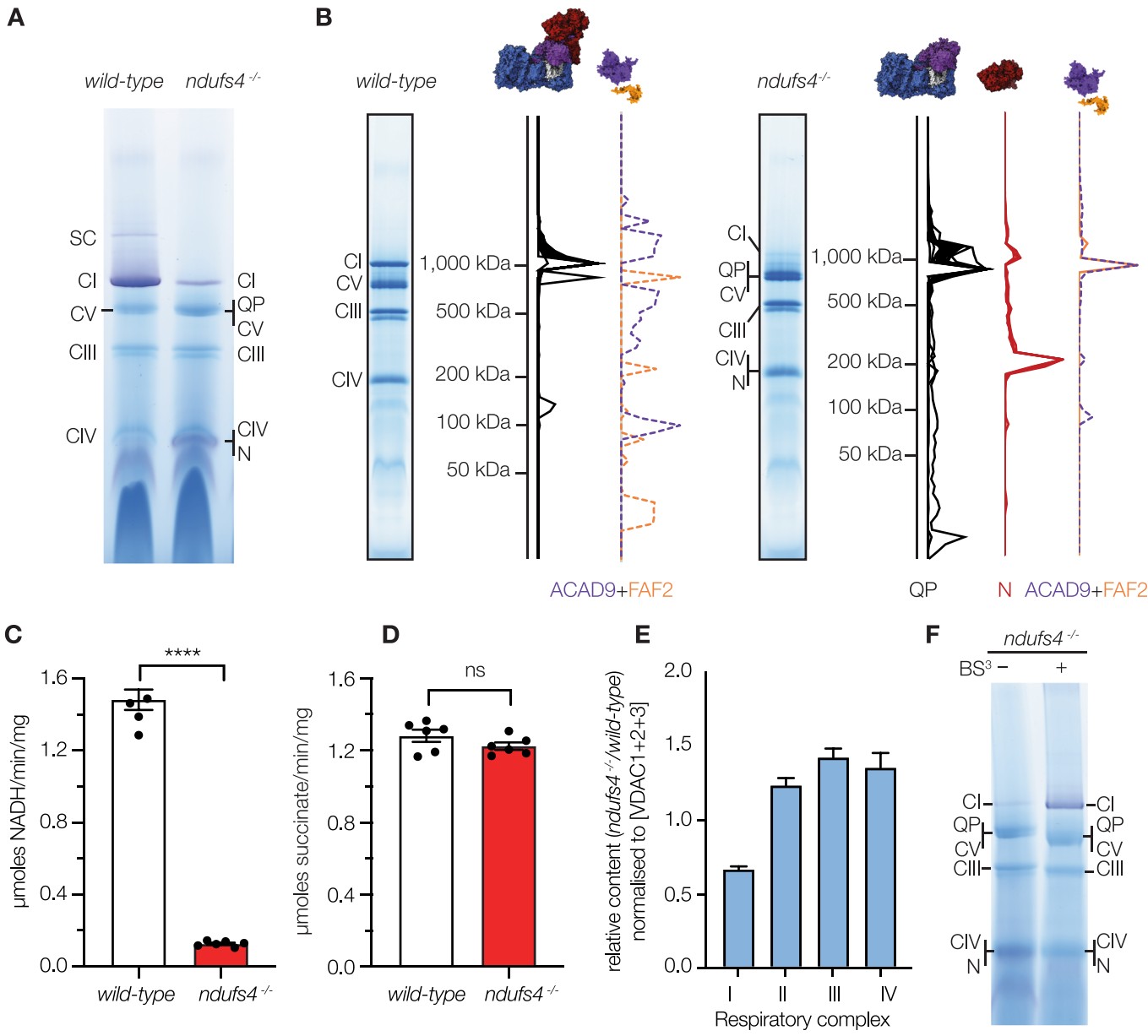

**Figure 2. Characterization of *ndufs4⁻/⁻* mouse mitochondrial membranes.**

(A) BN-PAGE analyses of heart mitochondrial membranes solubilized by 1% DDM and stained with the in-gel NADH-linked NBT assay with 30 µg of protein loaded on each lane. (B) Summary of the complexomic analyses of complex I subunits and assembly factors in wild-type and *ndufs4⁻/⁻* heart mitochondrial membranes. The complete datasets are shown in Fig. EV1. (C) The piericidin A-sensitive rates of NADH oxidation by wild-type and *ndufs4⁻/⁻* heart mitochondrial membranes using 200 µM NADH. Data are mean averages ± SEM ($n = 4$ or 6) analyzed with the Student's $t$-test (****$p < 0.0001$). (D) The atpenin A-sensitive rates of succinate oxidation using 5 mM succinate. Data are mean averages ± SEM ($n = 6$) analyzed with the Student's t-test. In C and D, each point represents a biological replicate (an individual membrane preparation from more than four mice). (E) Relative quantification of respiratory complexes I to IV in kidney mitochondrial membranes of *ndufs4⁻/⁻* mice compared to wild-type mice, calculated by normalizing the sum of MS peptide areas for the proteins in each complex to the sum of those for VDAC1, VDAC2, and VDAC3. Data are mean averages of technical replicates ± SEM with propagated error ($n = 3$ or 4). (F) BN-PAGE analyses of *ndufs4⁻/⁻* heart mitochondrial membranes solubilized by 1% DDM with and without cross-linking by BS³ (0.1 mM for 25 min) then stained with NADH-linked NBT. N, N-module of complex I (~200 kDa); QP, QP subcomplex of complex I (~800 kDa); SC, supercomplexes. Source data are available online for this figure.

mitochondrial matrix, but the deflavinated form of ACAD9 also forms part of the mitochondrial complex I intermediate assembly (MCIA) complex (containing NDUFAF1, ECSIT, and TMEM126B) that is expected to interact with newly translated subunit ND2 during assembly of the complex I QP module (Formosa et al, 2020). No intact MCIA was observed in our analyses (the other

components are observed comigrating at a higher apparent mass in the *ndufs4⁻/⁻* sample) and the comigration of ACAD9 with the ~800 kDa intermediate is therefore unexpected.

Kinetic measurements of the specific NADH:O₂ oxidoreductase activity (turnover of the CI-III-IV pathway, Fig. 2C) show that the rate of complex I-linked respiration by *ndufs4⁻/⁻* mouse heart

mitochondrial membranes is only ~10% of that of wild-type membranes. No corresponding difference was observed for the specific succinate:$O_2$ oxidoreductase activity (CII-III-IV, Fig. 2D), confirming that the loss of NADH:$O_2$ activity stems from a complex I defect. It could indicate a lower complex I content, and/or a lower intrinsic activity of the *ndufs4$^{-/-}$* complex relative to the wild-type complex. The amount of intact complex I present (irrespective of its capability to reduce ubiquinone) is typically assessed by (i) semi-quantitative evaluation of the intensities of Coomassie or NBT-stained bands on BN-PAGE (precluded here by enzyme instability, see below), or (ii) quantitative determination of the rates of flavin-localized reactions that couple NADH oxidation to reduction of artificial electron acceptors such as hexaammineruthenium III (HAR) or hexacyanoferrate (FeCN). Here, we found that the kinetics of the flavin-localized reactions were drastically altered in *ndufs4$^{-/-}$* membranes (Fig. EV2): the $K_M$ value for HAR is decreased 120-fold relative to in the wild-type enzyme, and the $K_M$ values for NADH are increased threefold for both the NADH:HAR and NADH:FeCN reactions. The altered values are consistent with the proximity of NDUFS4 to the flavin site, and the drastically decreased $K_M$ value for HAR also suggests the possibility of a new HAR binding site opening up in the *ndufs4$^{-/-}$* variant. They also preclude flavin-site kinetics being used to determine the complex I content. Instead, we turned to mass spectrometry of whole membrane digests and performed unlabeled quantification relative to the sum of peptides from VDAC1, 2 and 3: the relative abundance of *ndufs4$^{-/-}$* complex I peptides was $66 \pm 2\%$ of that observed for wild-type membranes (Fig. 2E). Although NDUFA12 peptides were detected in both wild-type and *ndufs4$^{-/-}$* samples, fewer peptides were detected for *ndufs4$^{-/-}$* and no peptides were in common to all replicates, precluding quantification. The unlabeled quantification was carried out using membranes from kidney to minimize animal usage and, in the samples studied, the NADH:$O_2$ rates were $1.36 \pm 0.38$ and $0.38 \pm 0.03$ µmol NADH $mg^{-1} min^{-1}$ (S.D. $n = 8$) for wild-type and *ndufs4$^{-/-}$* membranes, respectively ($28 \pm 2\%$, matching the earlier value of $25 \pm 11\%$ from kidney tissue (Calvaruso et al, 2012)). Our data thus suggest that in our samples, the lower NADH:$O_2$ activity in kidney membranes is due to both decreased content and decreased intrinsic catalytic activity of the *ndufs4$^{-/-}$* enzyme, which we estimate to catalyze at ~40% of the wild-type rate. We note that previous studies have observed disparities in the activity of complex I in intact cells or tissues compared to isolated mitochondria (Kruse et al, 2008; Valsecchi et al, 2012), and so cannot discount the possibility that, due to our preparatory procedure, the activity we observe in membranes is an underestimation of the activity in vivo.

The N-module is hydrophilic, so if it dissociates from the complex then it would be expected to be lost in the supernatant following collection of the membranes by centrifugation—yet we see it in mass spectrometry analyses of our membrane preparations. This observation suggests that, as suggested previously (Adjobo-Hermans et al, 2020), the N-module can remain attached to the QP subcomplex in vivo and in membrane preparations but readily dissociates during BN-PAGE. To test this proposal, *ndufs4$^{-/-}$* membranes were treated with the cross-linker bis(sulfosuccinimidyl)suberate (BS$^3$, which links nearby lysine residues to one another) before BN-PAGE, substantially increasing the amount of intact complex I observed (Fig. 2F). Note, however, that BS$^3$ treatment was deleterious for catalysis, decreasing the rate of

NADH:decylubiquinone oxidoreduction to $9.1 \pm 1.2\%$ of the control rate when added at 2.5 mM to DDM-solubilized bovine heart membranes, which we attribute to effects on catalytic enzyme motions and/or substrate access.

Together, our results suggest that lack of NDUFS4 destabilizes the association of the N-module to the *ndufs4$^{-/-}$* complex, but that the N-module remains bound in vivo and is lost during BN-PAGE analyses.

## Purification of *ndufs4$^{-/-}$* complex I and initial cryo-EM study of the kidney enzyme

Complex I from *ndufs4$^{-/-}$* mice was isolated using the same method as for the wild-type complex (Agip et al, 2018), by solubilization with DDM, ion-exchange and size-exclusion chromatography. As expected, the yield was lower, but the complex was purified successfully from both heart and kidney membranes with the same apparent molecular mass as for the wild-type enzyme (Fig. 3A,B). Kinetic measurements were then compared for the *ndufs4$^{-/-}$* and wild-type enzymes, using both heart membranes (Fig. 3C) and purified heart enzymes (Fig. 3D). The rates of NADH:ubiquinone oxidoreduction (NADH:$O_2$ and NADH:DQ reactions, respectively) were much lower for the *ndufs4$^{-/-}$* variant (<10% of the wild-type activity). Flavin-site assays using artificial electron acceptors (NADH:FeCN, HAR, and APAD$^+$) also showed decreased activities for *ndufs4$^{-/-}$* relative to wild-type, although the relative rates of each reaction varied. The rate of $H_2O_2$ production from the reaction of the NADH-reduced flavin with $O_2$ (determined using Amplex Red to detect both the $H_2O_2$ produced directly and the $H_2O_2$ produced by spontaneous dismutation of superoxide) (Kussmaul and Hirst, 2006) also decreased for *ndufs4$^{-/-}$* relative to wild-type complex I. As the purified *ndufs4$^{-/-}$* complex I was determined to contain $0.93 \pm 0.02$ molecules of FMN per 1 MDa of protein (close to the expected 1 FMN per MDa), and the thermal stability of the bound FMN was not altered substantially (Fig. 3E), it appears that the decreased flavin-site activities reflect a change in the catalytic properties of the site itself (rather than a decreased content of FMN). Finally, mass spectrometry analyses on the enzyme from *ndufs4$^{-/-}$* heart membranes showed that two additional complex I-linked proteins, ACAD9 and NDUFAF2, were co-purified with it (Fig. EV3; Appendix Table S1), with a substantial band visible on SDS-PAGE where both ACAD9 and a related protein, ACADVL, were detected. ACADVL is also required for β-oxidation (McAndrew et al, 2008) but it has not previously been reported to associate with complex I, or to play a role in respiratory complex assembly. ACAD9 and ACADVL both bind FAD, and a substoichiometric level of $0.35 \pm 0.01$ FAD was detected per MDa of protein. Mass spectrometry analyses of the wild-type enzyme only detected NDUFAF2 with poor quality peptide data and did not detect ACAD9, but a matching band on SDS PAGE was identified as ACADVL (Fig. EV3; Appendix Table S2).

For preliminary structure evaluation, complex I from *ndufs4$^{-/-}$* mouse kidney was subjected to cryo-EM imaging (by working with the enzyme from kidney as well as from heart we were able to decrease the number of mice required). No cross-linker treatment was used. The enzyme was frozen onto PEGylated UltrAuFoil R0.6/1 gold grids (Meyerson et al, 2014; Blaza et al, 2018) and imaged with an FEI Falcon III detector in integrating mode. Around 150,000 particles were picked, but 2D and 3D classification using cryoSPARC (Punjani et al, 2017) eliminated most of them because they did not resemble intact complex I, leading eventually to a

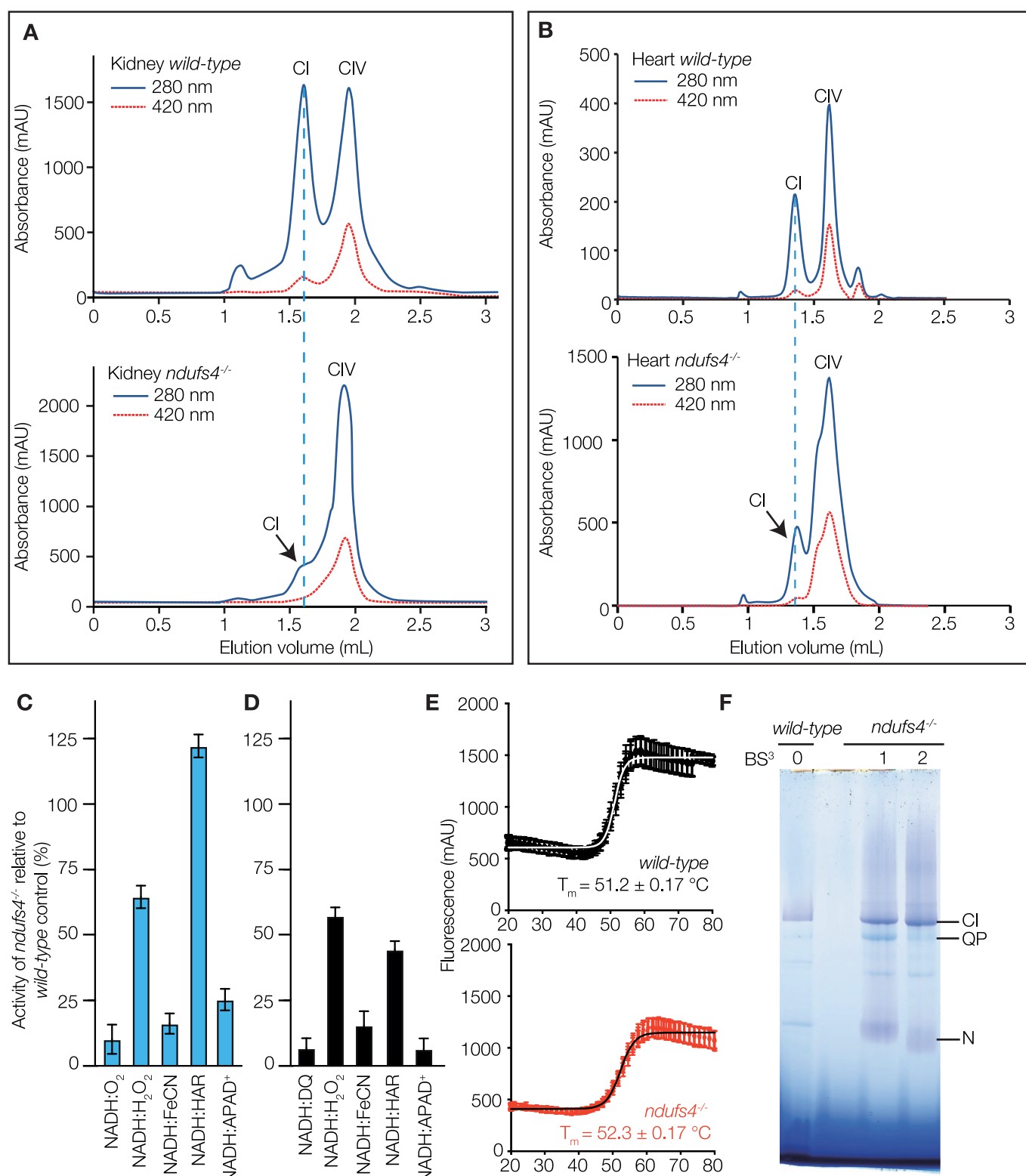

reconstruction from only 7563 particles with an estimated resolution of 6.2 Å (Appendix Fig. S1 and Appendix Table S3) [Electron Microscopy Data (EMD): 16514]. Despite the low yield of intact complex I particles and low final resolution, particularly for the N-module, the map was sufficient to observe a mature complex

I structure, which appeared to lack density for the NDUFS4 subunit, as well as for subunit NDUFS6 (Fig. EV4). As the protein eluting from the size exclusion column has a mass consistent with intact complex I (Fig. 3A), damage at the air-water interface during grid freezing probably causes the more fragile

**Figure 3.** Characterization of purified *ndufs4*⁻/⁻ complex I.

Size exclusion chromatography of complex I from (**A**) kidney and (**B**) heart mitochondrial membranes. The absorbance at 280 nm is in dark blue and at 420 nm in red. The peak elution volume of wild-type complex I is marked with a in dashed line. The peak volumes for complex I elution vary (~1.35 in A *vs.* ~1.6 mL in B) due to use of different columns (of the same type). (**C**) Rates of kinetic reactions for *ndufs4*⁻/⁻ heart mitochondrial membranes as a proportion of the wild-type rates (mean ± S.E.M., *n* ≥ 3 technical replicates). (**D**) Rates of kinetic reactions for purified *ndufs4*⁻/⁻ heart complex I as a proportion of the wild-type rates (mean ± S.E.M., *n* ≥ 3 technical replicates). (**E**) Loss of FMN from complex I purified from wild-type and *ndufs4*⁻/⁻ heart as the temperature is increased, measured as the fluorescence of the free flavin (mean ± S.E.M., *n* = 3 technical replicates). (**F**) The effect of BS³ cross-linking on BN-PAGE of the enzymes purified from heart; wild-type enzyme was not cross-linked (BS³ 0); *ndufs4*⁻/⁻ complex I was cross-linked with 0.25 mM BS³ during solubilization from the membranes, and then run as prepared (BS³ 1); *ndufs4*⁻/⁻ complex I was reacted again with 2.5 mM BS³ after size exclusion chromatography (BS³ 2). The second reaction further increased the stability of the enzyme (compare to the untreated sample in Fig. 2F). Source data are available online for this figure.

*ndufs4*⁻/⁻ complex to fragment, much like during BN-PAGE, leading to loss of most of the intact particles.

## Cryo-EM structures of *ndufs4*⁻/⁻ complex I from mouse heart mitochondria

To stabilize the complex during grid freezing and improve the yield of intact particles, the BS³ cross-linking procedure tested on membranes was extended to the purified enzyme. BS³ was applied twice, once during membrane solubilization and once immediately before the enzyme was added to the grids, a procedure that enabled a substantial proportion of the complex to survive BN-PAGE analysis (Fig. 3F). The cross-linked complex was frozen onto PEGylated UltrAuFoil R0.6/1 gold grids and 7309 micrographs from a single grid were collected on an FEI Titan Krios microscope fitted with a Gatan Quantum K3 Summit detector with energy filter (Appendix Fig. S2, Appendix Table S3). A low (7.1 Å)-resolution class, lacking clear density for the N-module, was found that corresponds to the shape predicted for the ~800 kDa QP subcomplex identified in complexome analyses, and the reconstruction of the intact complex I achieved a global resolution of 2.9 Å (Appendix Fig. S2). A survey of lysine densities failed to identify any continuous features from Lys-Lys cross-links, suggesting the enzyme is stabilized by a mixture of low-occupancy links. As the map for the intact enzyme showed clear signs of heterogeneity, the intact particles were then subjected to focused classification using cryoSPARC, resulting in three major classes (Appendix Figs. S3, S4). The class 1 map exhibited a large low-resolution protein density adjacent to subunit ND1, the class 2 map (70% of the particles) resembled the consensus map, and the class 3 map exhibited an additional protein density on the heel of the complex, where subunit NDUFA12 usually resides. All three classes were compared by map-to-map correlation to the mouse wild-type active and deactive resting states (EMD: 11377 (Bridges et al, 2020) and EMD: 11810 (Yin et al, 2021)) and all three were found to resemble the active state more closely than the deactive state (Appendix Table S4). Furthermore, all the specific hallmarks of the active state are present in all three maps, including the relative orientation of NDUFA5 and NDUFA10, the lack of π-bulge in ND6-TMH3, and the ordered states of loops in NDUFS2, NDUFA9, ND3 and ND1 (Agip et al, 2018; Chung et al, 2022a).

## Deletion of NDUFS4 weakens the attachment of the N-module

Comparing the class 2 model of *ndufs4*⁻/⁻ complex I to the model of the active state of wild-type mouse complex I (we use PDB: 6ZR2 (Bridges et al, 2020) as our reference structure throughout) shows that deletion of the NDUFS4 subunit leaves a deep groove on the surface of the hydrophilic domain (Fig. 4A). In the consensus class 2 map, the density for the N-module is substantially lower in resolution than the density for the rest of the enzyme but when the N-module is locally refined its resolution improves to the same level (Fig. 4B; Appendix Fig. S4A-B). The resultant N-module model (subunits NDUFV1, NDUFV2, NDUFV3, NDUFS1 and NDUFA2) has an all-atom RMSD of 1.20 Å against the equivalent N-module of the wild-type (NDUFS4-containing) reference model, and the density for the FMN is clear, in its expected location. Thus, the structure of the N-module itself is closely conserved in structure between the wild-type and *ndufs4*⁻/⁻ enzymes, but in *ndufs4*⁻/⁻ the module is not rigidly fixed to the rest of the enzyme and adopts a distribution of positions relative to it (Fig. 4C and Movie EV1). Finally, the NDUFA9 subunit drops slightly towards the membrane, relative to its position in the wild-type reference structure (Fig. 4D and Movie EV1), due to loss of stabilizing contacts from the NDUFA9 N-terminus to subunits NDUFS4, NDUFS6 and NDUFA12, at a crucial junction between all four subunits (Fig. 1C).

Within the *ndufs4*⁻/⁻ N-module, residues 61-68 of subunit NDUFV3 that are adjacent to NDUFS4 and resolved in the wild-type enzyme, remain disordered even after local refinement. Subunits NDUFS2, NDUFS3, NDUFS8, and NDUFA5, which are adjacent to the N-module in the QP subcomplex and usually interact with NDUSF4, are not changed by its absence. The C-terminus of NDUFA6 is unchanged in its position on the N-module, but shifted relative to the QP subcomplex. There is no density observed for subunit NDUFA12, or for the N-terminal regions of NDUFS6 (residues 1–39), NDUFA7 (residues 1–35), and NDUFA9 (residues 1–17) that are well ordered in the wild-type reference structure (Fig. EV5). The proximity of these features in the wild-type enzyme (Fig. 4E) suggests a cascade of effects, in which the absence of NDUFS4 affects the N-terminus of NDUFA9 and the C-terminal loop of NDUFA12, which is normally overlaid by the N-terminal domain of NDUFS6, and loss of NDUFA12 propagates to the adjacent N-terminus of NDUFA7. Loss of NDUFA12 is consistent with our mass spectrometry analyses (Figs. EV1 and EV3), as well as with previous proteomic studies of *ndufs4*⁻/⁻ mouse tissues, *ndufs4*⁻/⁻ mouse embryonic fibroblasts, and NDUFS4-mutated human cells (Adjobo-Hermans et al, 2020; Stroud et al, 2016). The C-terminal domain of NDUFS6 and the rest of NDUFA7 appear otherwise unaffected structurally, with the Zn cofactor of NDUFS6 in its expected location. However, like the N-module, these two subunits were not found to co-migrate with the rest of the *ndufs4*⁻/⁻ complex in complexome analyses (Figs. 2B and EV1) suggesting they are also only weakly bound.

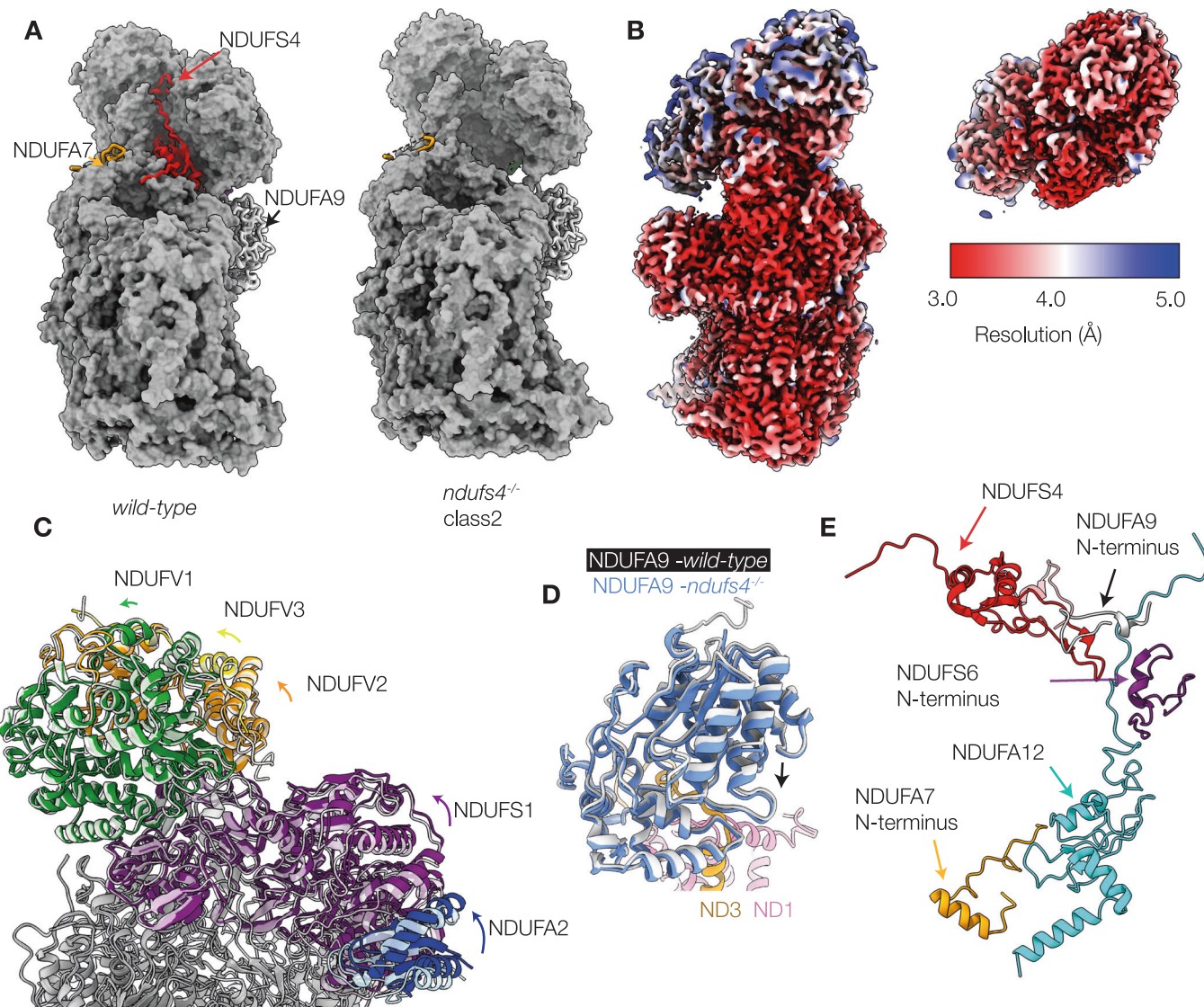

**Figure 4. Structural differences between *ndufs4⁻/⁻* heart complex I (class 2) and the wild-type enzyme.**

(**A**) View of the active wild-type reference model (PDB: 6ZR2) and the *ndufs4⁻/⁻* heart class 2 model (PDB: 8CA3) showing the crevice left by the absence of NDUFS4. Red, NDUFS4; gold, NDUFA7; white, NDUFA9. (**B**) Local resolutions of the consensus map of *ndufs4⁻/⁻* heart complex I (class 2, EMD-16516) and the locally refined N-module map (class 2, EMD-16517). (**C**) Overlay of the PDB models for the *ndufs4⁻/⁻* (PDB: 8CA3) and wild-type (PDB: 6ZR2) enzymes. The wild-type model is shown in semi-transparent, and the rotation of the colored subunits of the N-module between the two enzymes is observed upon global model alignment. Green, NDUFV1; yellow, NDUFV3; purple, NDUFS1; blue, NDUFA2. (**D**) Overlay of the PDB models for the *ndufs4⁻/⁻* (PDB: 8CA3) and wild-type (PDB: 6ZR2) enzymes focusing on the NDUFA9 subunit. The wild-type model is shown in white, and subunits of the *ndufs4⁻/⁻* enzyme are in color. With the models aligned on core subunit ND1, NDUFA9 drops slightly in the *ndufs4⁻/⁻* enzyme (see arrow). (**E**) The regions of the active wild-type enzyme which appear disordered in *ndufs4⁻/⁻* maps.

## Association of assembly factor NDUFAF2 with *ndufs4⁻/⁻* complex I

The structure of the class 3 *ndufs4⁻/⁻* complex I matches that from class 2 closely, except that assembly factor NDUFAF2 is bound in the place occupied by subunit NDUFA12 in the wild-type complex (Fig. 5A), and the whole of the adjacent NDUFS6 subunit is absent, including the Zn-containing C-terminal domain present in class 2 (see Table 1 for a comparison of complex I compositions). The presence of NDUFAF2 is consistent with our complexomic analyses and mass spectrometry on the isolated enzyme, although the particle classification data suggest that only ~14% of the

population contains NDUFAF2, while the other ~86% (classes 1 and 2) contain NDUFS6 instead: NDUFS6 and NDUFAF2 were not observed together in any class. The location of NDUFAF2 overlaps with that of NDUFA12 (in the wild-type enzyme) through a common three-strand β-sheet and adjacent short helix; however, NDUFAF2 also contains a long 'diagonal' helix that is not present in NDUFA12, which crosses the region occupied by the small central helix of NDUFS6 in the wild-type enzyme (Figs. 5B and 1C). A 35-residue unresolved stretch of NDUFAF2 is followed by a C-terminal β-hairpin, which interacts with subunit NDUFS1 (Fig. 5B,C). Our experimental structure of NDUFAF2 matches well with the AlphaFold prediction (Jumper et al, 2021), with

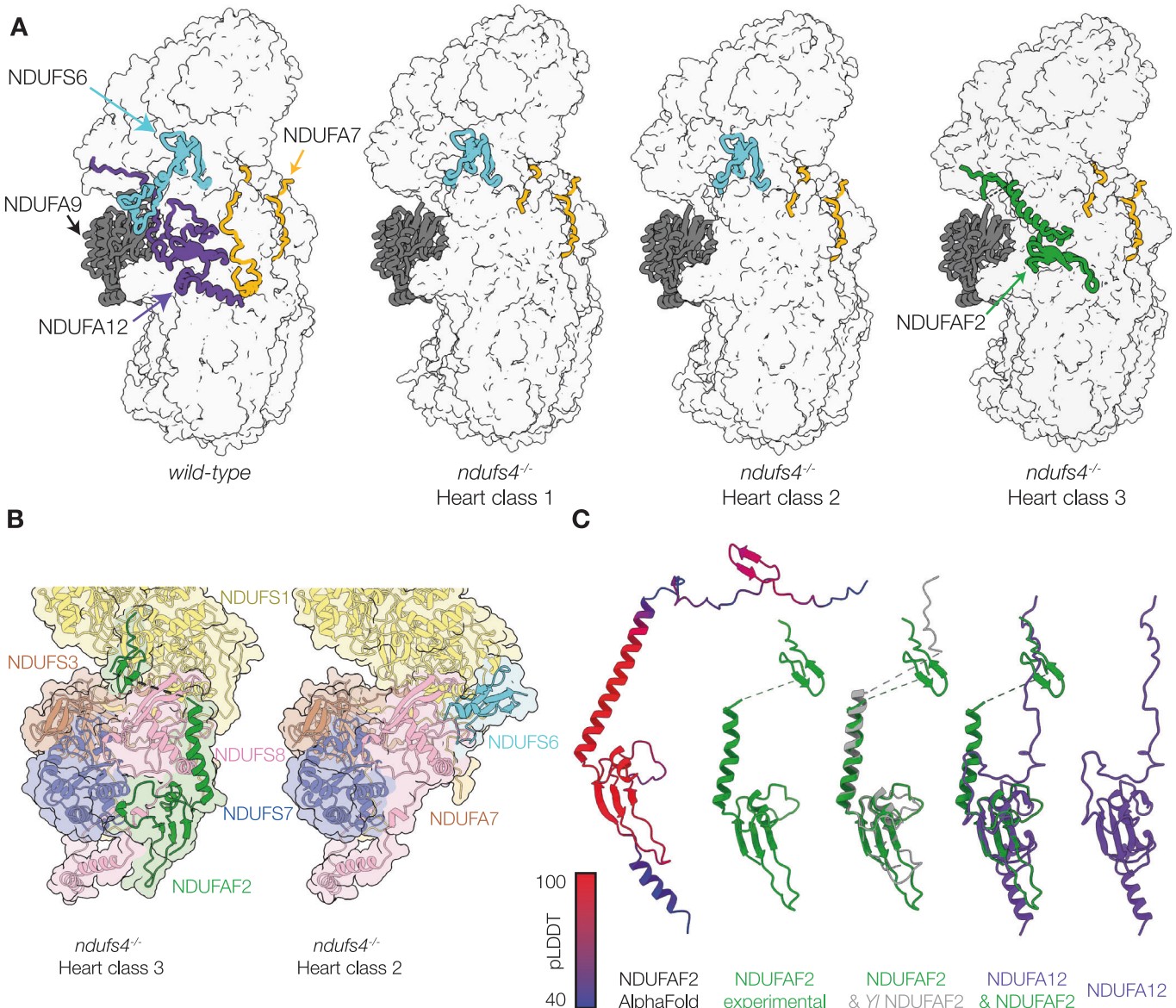

**Figure 5. NDUFAF2 binding in *ndufs4*−/− class 3 and comparison of the NDUFAF2- and NDUFS6-bound *ndufs4*−/− structures.**

(**A**) Complex I models for the wild-type (left, PDB: 6ZR2) and *ndufs4*−/− classes 1 (PDB: 8C2S), 2 (PDB: 8CA3) and 3 (PDB: 8CA5) shown in surface view for the majority of the complex. Cartoon view is shown for subunits NDUFS6 (teal), NDUFA7 (gold), NDUFA9 (gray), NDUFA12 (purple), and NDUFAF2 (green). Protein compositions are also summarized in Table 1. (**B**) Relationships between NDUFAF2 and surrounding subunits in class 3, and the equivalent view for class 2. An equivalent view for the wild-type complex is shown in Fig. 1C. (**C**) Comparison of the AlphaFold-predicted model for mouse NDUFAF2 (AF-Q59J78), experimental models for NDUFAF2 from *ndufs4*−/− class 1 (PDB: 8C2S, green) and *Y. lipolytica ndufs6*−/− (PDB: 7RFQ, gray) and subunit NDUFA12 from wild-type mouse (PDB: 6ZR2, purple). The AlphaFold pLDDT score indicates the confidence of the model with values above 90 indicating very high confidence and values below 70 low confidence; regions with values below 50 may be intrinsically disordered.

regions predicted but not observed in our structure (an N-terminal α-helix, loop between the diagonal helix and the C-terminal β-hairpin, and a C-terminal coil) only predicted with low confidence, and with AlphaFold scores indicating they may be disordered (Fig. 5C). The location and fold of NDUFAF2, including regions that are not well-ordered, also matches well to that observed in the model of the ΔNDUFS6-complex I from *Y. lipolytica* (Parey et al, 2019) (Fig. 5C) except that the *Y. lipolytica* model lacks the β-hairpin. Finally, the low-resolution map from *ndufs4*−/− kidney matches best to the heart class 3 map, with density for the

NDUFAF2 diagonal helix clearly visible, and no density observed for NDUFS6 (Fig. EV5).

## ACAD association with *ndufs4*−/− complex I

The 12% of particles in *ndufs4*−/− class 1 were found to have an additional large low-resolution density on the heel of the complex. Particle subtraction of this region followed by local refinement with C2 symmetry (Fig. 6A and Appendix Fig. S3) produced a 4.2-Å resolution map for the additional density, which was visually

**Table 1. Comparison of the subunit compositions of mouse and *Y. lipolytica* complex I models.**

| | | | Subunits | | | | | |
|---|---|---|---|---|---|---|---|---|
| | | PDB code | N-module | NDUFS4 | NDUFS6 | NDUFA12 | NDUFAF2 | ACADVL |
| Mouse heart | *ndufs4*⁻/⁻ class 1 | 8C2S | + | | + | | | + |
| | *ndufs4*⁻/⁻ class 2 | 8CA3 | + | | + | | | |
| | *ndufs4*⁻/⁻ class 3 | 8CA5 | + | | | | + | |
| | wild-type | 6ZR2 | + | + | + | + | | |
| *Y. lipolytica* | *ndufs4*⁻/⁻ | 6RFS | + | | + | + | | |
| | *ndufs6*⁻/⁻ | 6RFQ | + | + | | | + | |
| | wild-type | 6YJ4 | + | + | + | + | | |

identified as a dimer of an ACAD-family protein. Mass spectrometry on purified heart *ndufs4*⁻/⁻ complex I (prepared without cross-linking) identified two candidates, ACAD9 and ACADVL, both with mass ~65 kDa (Fig. EV3). Although our complexome analyses of DDM-solubilized *ndufs4*⁻/⁻ heart membranes (Figs. 2B and EV1) revealed co-migration of ACAD9 and the ~800 kDa QP subcomplex (but not co-migration of ACADVL), our MALDI-TOF and LC/MS analyses both contained substantially more peptides for ACADVL (Appendix Table S1). Considering the sequence similarity between the two proteins, these data suggest ACADVL is present in greater abundance than ACAD9. We thus chose to model the density as a dimer of ACADVL (Fig. 6B), and a survey of positions where bulky residues (Tyr, Trp, and Phe) vary between the two subsequently confirmed it as the major species.

The resolution achieved was sufficient to model the two FAD cofactors in the ACAD dimer (Fig. 6B), accounting for the ~0.3 FAD per MDa protein observed in the purified complex (two FAD per dimer bound to 12% of particles). Overall, the density corresponds closely to the model of human ACADVL solved by X-ray crystallography (PDB: 3B96) (McAndrew et al, 2008), but with an extra α-helical region (residues 490-518) resolved. This region is inserted into the micelle of our cryo-EM density and braces the dimeric ACAD against the hydrophobic heel of the complex (Fig. 6C), with one helix interacting with subunit ND1 and the other approaching subunit NDUFA3. However, the position of the ACAD dimer is flexible relative to the complex I and their interface is not well resolved. Comparison of the local structures of *ndufs4*⁻/⁻ complex I subunits (ND1, NDUFA3, NDUFA8, and NDUFA13) with the wild-type reference model revealed no differences, including in the closest interacting sidechains, and no substantive differences were observed between class 1 (ACAD-containing) and class 2 (no ACAD) either. Finally, as the ACAD bound to class 1 does not overlap spatially with the NDUFAF2 bound to class 3 we do not expect them to be mutually exclusive, so it is likely that the expected fourth class that contains both has simply been missed due to its low population (predicted ~1250 particles or 1.8%).

The location of ACAD binding in our map (Fig. 6A) is surprising: ACAD proteins are known to be membrane-associated and located in the mitochondrial matrix (McAndrew et al, 2008), but in our structure the ACAD dimer appears to have 'slipped' around, into a position that would be occupied by the lipid tails of a membrane bilayer. Furthermore, the assembly factor ACAD9, as part of the MCIA complex, is expected to be deflavinated and to interact with subunit ND2 (Formosa et al, 2020), which is inconsistent with our flavinated ACADVL interacting with subunit ND1. No density to suggest the presence of a similarly bound ACAD could be identified in the maps from the *ndufs4*⁻/⁻ kidney enzyme, which was not treated with a cross-linker, or in any of our structural data on wild-type mouse complex I (Agip et al, 2018; Bridges et al, 2020; Grba et al, 2022), even though the abundance of ACAD proteins in *ndufs4*⁻/⁻ and wild-type mitochondria are similar (Fig. 6D). Therefore, we conclude the ACAD dimer is artefactually bound, and that its observed binding mode is not physiologically relevant or involved in enzyme assembly. The ACAD dimer is stabilized in the *ndufs4*⁻/⁻ enzyme by the cross-linking procedure that was applied to stabilize the N-module: although we do not observe connecting density, the positions of Lys sidechains on ACADVL and NDUFA8 are consistent with reaction with the BS³ cross-linker (Fig. 6E).

## Structure of complex I from *ndufs4*⁻/⁻ mouse contains bound ubiquinone

Continuous density resembling that expected for ubiquinone-9 was identified along the length of the ubiquinone-binding channel in the class 2 *ndufs4*⁻/⁻ enzyme (Appendix Fig. S5A). The density feature observed is substantially longer than the BS³ cross-linker molecule, and so we conclude it originates from bound ubiquinone, even though the density for the ubiquinone headgroup is poorly defined, precluding a confident model being built. Similar but less contiguous densities were also observed in class 3, whereas the densities observed in class 1, as well as in the reference map from the wild-type complex (EMD: 11377 (Bridges et al, 2020)), were fragmented and could not be interpreted. Although the class 2 *ndufs4*⁻/⁻ ubiquinone headgroup is not resolved, the residues around its binding site match closely to the residues in the wild-type reference model (Appendix Fig. S5B). Comparison to the models for mouse complex with inhibitors bound (Grba et al, 2022; Bridges et al, 2020), and to other mammalian complexes with ubiquinone-10 bound in the active site (Chung et al, 2022b; Gu et al, 2022) (Appendix Fig. S5C), show that the isoprenoid chain overlays well with the location of other inhibitors and substrates modeled in the channel. Finally, our model provides no explanation for why our class 2 *ndufs4*⁻/⁻ map exhibits better density for ubiquinone-9 than the wild-type enzyme (Agip et al, 2018; Bridges et al, 2020), even at higher resolution. The difference may be due to an indirect effect of the absence of NDUFS4 or the BS³ cross-linker, perhaps restricting an enzyme motion required for ubiquinone

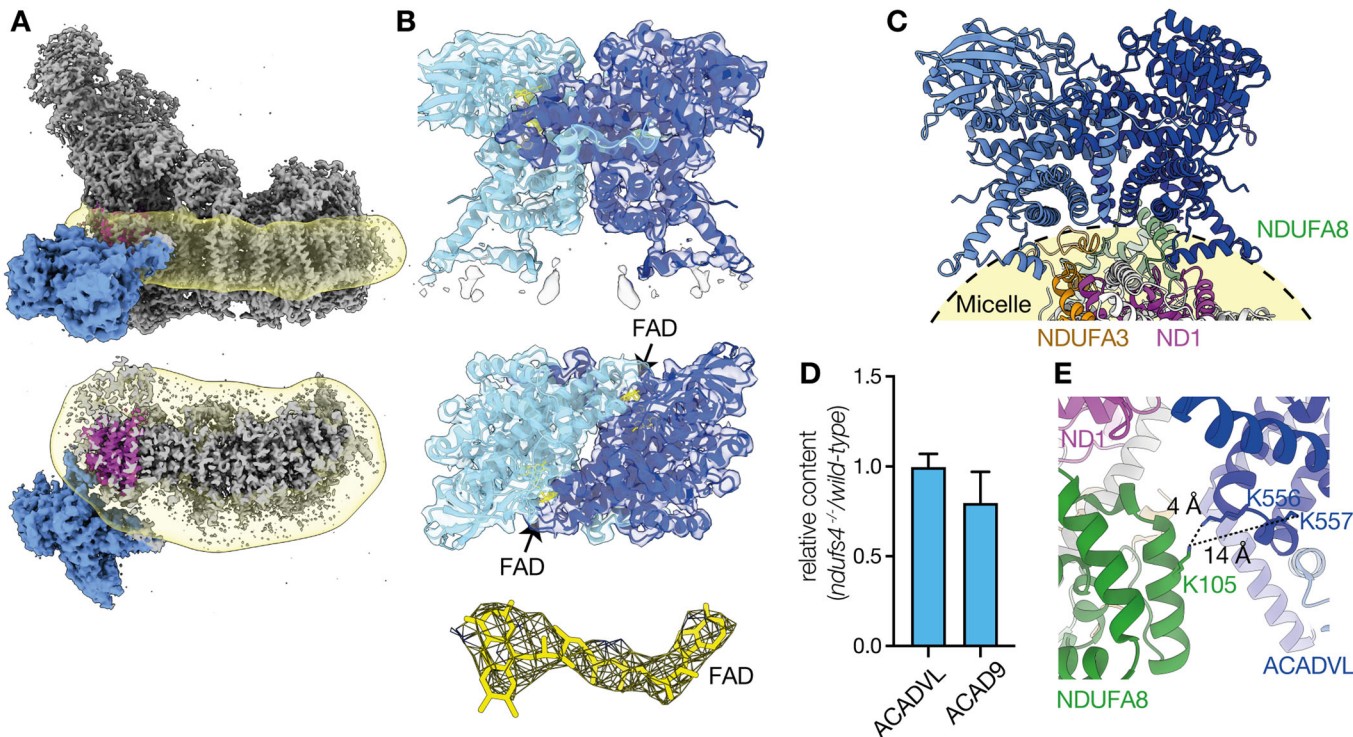

**Figure 6.  ACADVL association with purified *ndufs4*⁻/⁻ heart complex I.**

(**A**) Cryo-EM density map showing the location of the ACAD dimer (blue) attached to the detergent micelle (yellow) and next to subunit ND1 (purple). (**B**) The model for the ACADVL dimer and cryoEM density (at a threshold of 1.74) is shown with the location of the FAD cofactors highlighted in yellow; the FAD density is shown separately also (right, at a threshold of 1.74). (**C**) The dimeric model of ACADVL aligned to its position relative to complex I, within the detergent micelle. (**D**) Relative quantification of ACAD9 and ACADVL. The sum of peptide areas for each protein were normalized to the sum of peptide areas for VDAC1, VDAC 2, and VDAC3, and displayed as the ratio between wild-type and *ndufs4*⁻/⁻ kidney mitochondrial membranes. Data are mean averages of technical replicates ±SEM with propagated error ($n = 3$ or 4). (**E**) The short Lys-Lys distances between ACADVL and NDUFA8 that may be responsible for the stabilization of ACAD binding by crosslinking. Source data are available online for this figure.

exchange, consistent with the loss of catalytic activity in the *ndufs4*⁻/⁻ complex, and with further loss upon cross-linker treatment.

## Discussion

### Is *ndufs4*⁻/⁻ complex I unstable or incompletely assembled?

Purification and cryo-EM reconstruction of complex I from *ndufs4*⁻/⁻ mouse kidney and heart tissue yielded maps of an intact complex I, with the N-module bound, which contains almost all the subunits of the enzyme (including all the core-subunits) and exhibits a wild-type-like mass on BN-PAGE. The ~800 kDa subcomplex identified on BN-PAGE both here and previously (Adjobo-Hermans et al, 2020; Calvaruso et al, 2012) likely results from fragmentation of the intact complex under BN-PAGE conditions, which we mitigated by adding a cross-linker (Fig. 2F). The presence of an intact complex in mouse in vivo explains why the *ndufs4*⁻/⁻ variant is not embryonically lethal, although the amount of enzyme present is substantially diminished (by ~50%), as well as less stable than the wild-type enzyme. A similar conclusion, that the *ndufs4*⁻/⁻ enzyme is intact in vivo, was drawn

for *Y. lipolytica* complex I (Parey et al, 2019; Kahlhöfer et al, 2017), and has also been suggested previously for the mammalian complex (Adjobo-Hermans et al, 2020; Valsecchi et al, 2012; van de Wal et al, 2022).

The conclusion that the ~800 kDa subcomplex observed on BN-PAGE is thus an artefact of the analysis urges caution when using BN-PAGE to assess clinically identified complex I defects, where genetic variants that result in a less stable enzyme may be conflated with those that genuinely result in interrupted-assembly intermediates. However, as well as being unstable, *ndufs4*⁻/⁻ complex I is also incompletely assembled: although it contains the N-module, it lacks NDUFA12 as well as NDUFS4, nearby structural elements are disordered, and it exists in different stages of completion, with some complexes still containing NDUFAF2, and some with NDUFS6 missing or incompletely associated (Figs. 4 and 5).

### Loss of catalytic activity in the absence of NDUFS4

Our kinetic assays revealed substantially lower NADH:ubiquinone oxidoreductase activity for *ndufs4*⁻/⁻ relative to wild-type complex I (Fig. 3C,D), even when the decreased amount of enzyme present in membranes is taken into account. The reason for the lower activity, which was also observed for the *Y. lipolytica ndufs4*⁻/⁻ enzyme (Kahlhöfer et al, 2017), is obscure. The static structure of the flavin

NADH-oxidizing site does not appear to be changed by the absence of NDUFS4, but the kinetics of reactions normally localized only at the flavin site are altered (Figs. 3C,D and EV2) and therefore the dynamics of the site may be different. However, NADH oxidation is not rate limiting during turnover (Birrell et al, 2009) so the moderate decrease in NADH oxidation rate by the $ndufs4^{-/-}$ complex is unlikely to affect the overall rate substantially. The NDUFS4 C-terminus lies within 8 Å of the cluster 1 in the FeS chain (the 4Fe-4S cluster in NDUFV2 that accepts the electrons from the flavin), the β1-β2 loop within 12 Å of clusters 2, 3 and 4 (NDUFS1), and the β2-β3 loop within 12 Å of cluster 6 (the penultimate, 4Fe-4S cluster in NDUFS8) (Fig. 1D). Furthermore, the small drop in NDUFA9 towards the membrane in the $ndufs4^{-/-}$ variant expands a solvent-filled cavity that extends towards clusters 5 and 6, the two [4Fe-4S] clusters in NDUFS8, although neither becomes directly solvent accessible. It is possible that the altered kinetics (particularly the decreased $K_M$ value) for the NADH:HAR oxidoreduction reaction result from interaction of HAR at a new $ndufs4^{-/-}$-specific site in this region. In *Y. lipolytica* complex I, deletion of NDUFS4 perturbed the electron paramagnetic resonance spectra of reduced FeS cluster 1 (signal N3), 2 (signal N1b, [2Fe-2S] cluster in NDUFS1), and 6 (signal N4), consistent with their changed dielectric environment (Kahlhöfer et al, 2017). However, as for NADH oxidation, the rate of intramolecular electron transfer is considered fast (the FeS chain is reduced during steady-state catalysis in wild-type complex I) and so it is unlikely to become rate limiting. For *Y. lipolytica* $ndufs4^{-/-}$ the increased solvent accessibility of the clusters was proposed to increase electron leakage to $O_2$, then further oxidative damage to the enzyme in a vicious cycle (Kahlhöfer et al, 2017). Increased cellular ROS production, relative to wild-type, was reported in $ndufs4^{-/-}$ mouse fibroblasts (Valsecchi et al, 2013), but not in $ndufs4^{-/-}$ mouse heart (Chouchani et al, 2014), and here we did not observe any substantial increase in ROS production in either $ndufs4^{-/-}$ mitochondrial membranes or complex I (Fig. 3C,D), with the rate of $H_2O_2$ production remaining two orders of magnitude slower than the rate of turnover. Increased complex I-linked ROS in mitochondria may result from changes to the environment, as well as from the complex I molecule itself, as a result of increased reduction of the $NAD^+$ pool and/or increased local $O_2$ concentration (Kussmaul and Hirst, 2006).

The decreased catalytic activity may also result from the altered dynamics and conformational landscape that likely results from loss of the structural junction between NDUFS4, NDUFS6 and NDUFA12, which pins N-module subunit NDUFS1 to NDUFA9. The NDUFA9 C-terminal domain, buttressed by the NDUFS7 C-terminus at the base of the hydrophilic domain, makes tight interactions with specific phospholipids at the membrane interface, and may affect the membrane topology near the ubiquinone-binding site. A further effect on the local membrane structure may arise from the absence of the NDUFA12 amphipathic helix, which is not present in NDUFAF2 (Padavannil et al, 2022), and together these may change the ubiquinone-9/10 binding dynamics. It is possible that the same effect explains the unexpected retention of ubiquinone-9 in the $ndufs4^{-/-}$ enzyme (Appendix Fig. S5). Finally, conformational changes in the NDUFA9 C-terminal domain have been observed in the complex I active-to-deactive transition (Zhu et al, 2016), and the C-terminal peptide contacts the start of the ND3-TMH1-2 loop, a structural element central to deactivation

(Chung et al, 2022a) that may also change its conformation during catalysis (Cabrera-Orefice et al, 2018). We were, unusually, unable to find a class of particles that resembled the deactive state in our $ndufs4^{-/-}$ complex I, so these subtle changes may also alter the active-to-deactive transition in the $ndufs4^{-/-}$ enzyme. More complete understanding of the structural changes that occur during catalysis is required to further evaluate these proposals.

## Comparison with *Y. lipolytica* $ndufs4^{-/-}$ complex I

In the *Y. lipolytica* $ndufs4^{-/-}$ strain, complex I levels decreased to ~40% of wild type, similar to here, but the subunits of the N-module plus NDUFS6, NDUFA12, and NDUFAF2 were all found to stay associated with the intact complex on BN-PAGE (Kahlhöfer et al, 2017). Subsequently, a 4 Å-resolution cryo-EM map (PDB: 6RFS) showed that NDUFS6 and NDUFA12 were bound as normal, but no density for assembly factor NDUFAF2 was observed (Parey et al, 2019) (Table 1). Thus, unlike the mouse $ndufs4^{-/-}$ enzyme, the *Y. lipolytica* $ndufs4^{-/-}$ complex has completed assembly of its hydrophilic domain without NDUFS4. Conversely, the mouse $ndufs4^{-/-}$ enzyme contains a mature membrane domain, but the *Y. lipolytica* $ndufs4^{-/-}$ complex does not: it lacks density for the N-terminal half of NDUFA11 and the C-terminus of ND5 (part of the transverse helix and TMH16), and has lost structural integrity for ND6-TMH4 and adjacent loops, as well as for the N-terminus of NDUFS2. NDUFA11 has been reported to associate in the final stages of complex I assembly (Andrews et al, 2013; Guerrero-Castillo et al, 2017) when the N-module joins the complex, so it is possible that assembly of NDUFA11 onto the *Y. lipolytica* $ndufs4^{-/-}$ complex stalls without NDUFS4. We note that the structural changes at/around *Y. lipolytica* NDUFA11 resemble those observed in the 'slack' state of mammalian complex I (Zhu et al, 2016; Chung et al, 2022b), so may also be caused by instability of the $ndufs4^{-/-}$ complex, either during purification or cryo-EM grid freezing. Finally, like our mouse $ndufs4^{-/-}$ class 3 structure, the structure of *Y. lipolytica* $ndufs6^{-/-}$ complex I (PDB: 6RFQ) (Parey et al, 2019) contains NDUFAF2 instead of NDUFA12, consistent with a helix in NDUFAF2 occupying the same space as the loop in NDUFS6 that connects its two domains together, so that both cannot bind together.

## Assembly pathway

Work on the assembly of both mammalian and yeast complexes I has led to a model in which the assembly factor NDUFAF2 recruits the N-module onto the nascent complex; NDUFS4 (and perhaps NDUFS6) may also be required to stabilise the N-module, before NDUFAF2 is displaced by NDUFS6 and NDUFA12 (perhaps as a single unit) to complete the enzyme (Lazarou et al, 2007; Stroud et al, 2016; Parey et al, 2019; Adjobo-Hermans et al, 2020; Pereira et al, 2013; Guerrero-Castillo et al, 2017). Comigration of NDUFAF2 with complex I in BN-PAGE analyses of NDUFA12-patient/knockout cells (Adjobo-Hermans et al, 2020; Stroud et al, 2016) has indicated NDUFA12 is essential for the displacement step, but not for N-module attachment. Although the overall catalytically competent complex I population present in the patient-derived cell line contains NDUFS4, NDUFAF2, and the N-module (Adjobo-Hermans et al, 2020), in light of our structural

data it is not known whether they are all present in a single homogenous population, or whether different protein combinations are present in different subpopulations. Finally, NDUFAF2, NDUFS4, NDUFS6, and NDUFA12 have also been considered an 'assembly checkpoint', blocking completion of the hydrophilic arm until the rest of the complex is complete (Padavannil et al, 2022).

Our structures of mouse *ndufs4⁻/⁻* complex I show that the N-module can be stabilized on the nascent complex by NDUFAF2 (without NDUFS4, NDUFS6 or NDUFA12, in class 3), or by the N-terminal domain of NDUFS6 (without NDUFS4, NDUFA12 or NDUFAF2, in class 2). We do not observe the N-module bound alone. However, while the N-terminal domain of NDUFS6 binds between N-module subunits NDUFS1 and NDUFV2 and QP-subcomplex subunits NDUFS2 and NDUFS8 (Fig. 5B), clearly stabilizing attachment of the N-module, the stabilizing effects of NDUFAF2 are unclear. Its N-terminal domain (modeled from Arg20 to Tyr106) is bound (predominantly) to subunits NDUFS7, ND1, and NDUFS8 on the heel of the complex, whereas its C-terminal domain (modeled from Glu141 to Glu161) is located on NDUFS1 in the N-module (Fig. 5B). There is no sturdy connection between them, since residues 107 to 140 are disordered—and indeed this stretch of the sequence is unstructured in the AlphaFold prediction AF-Q59J78 (Jumper et al, 2021). Therefore, NDUFAF2 does not form a stabilizing bridge to attach the N-module. Instead, we propose that it acts as a 'lasso' to capture it and guide it onto the complex: with its N-terminal domain firmly attached to the heel, its intrinsically disordered region allows the C-terminus to search the surrounding space, capture the N-module, and then restrict its freedom to promote docking (Fig. 7). Our proposal explains how NDUFAF2 increases the efficiency of (but is not essential for (Ogilvie et al, 2005)) N-module attachment and complex maturation, and why it cannot be substituted by NDUFA12, which subsequently replaces it on the mature wild-type complex but lacks the flexibly linked β-hairpin anchor to NDUFS1. Thus, our class 3 NDUFAF2-bound structure likely represents the first N-module-containing species on the assembly pathway.

Our structures show that the three subunits required to complete the mature protein (NDUFS4, NDUFS6, and NDUFA12) all overlap with NDUFAF2 and therefore likely contribute to destabilising and displacing it, suggesting they all bind only after the N-module has been recruited. The most substantial overlap is with NDUFA12, consistent with its importance in the displacement step (Adjobo-Hermans et al, 2020; Stroud et al, 2016). The β2-β3 hairpin-loop (residues 62-77) of NDUFS4, which is probably inserted from the front of the hydrophilic domain, to pass under the N-module and connect to NDUFS6, NDUFA12 and NDUFA9 (Fig. 1C), displaces a short section of the C-terminus of NDUFAF2. This section is ordered in our class 3 *ndufs4⁻/⁻* structure, but disordered in the structure of *Y. lipolytica ndufs6⁻/⁻* complex I that contains both NDUFS4 and NDUFAF2 (Parey et al, 2019). The NDUFA12 N-terminal domain replaces the NDUFAF2 N-terminal domain on the heel, and its C-terminus replaces the NDUFAF2 C-terminus on NDUFS1 (Fig. 5C). The connecting loop, which is ordered in NDUFA12 and interacts with the NDUFS4 β2-β3 loop and NDUFS6, is trapped under the NDUFS6 C-terminal domain in the mature complex (Fig. 1C), which must thus bind either after or along with it. Our class 2 *ndufs4⁻/⁻* structure, representing the majority of the particles in the heart complex, and containing the

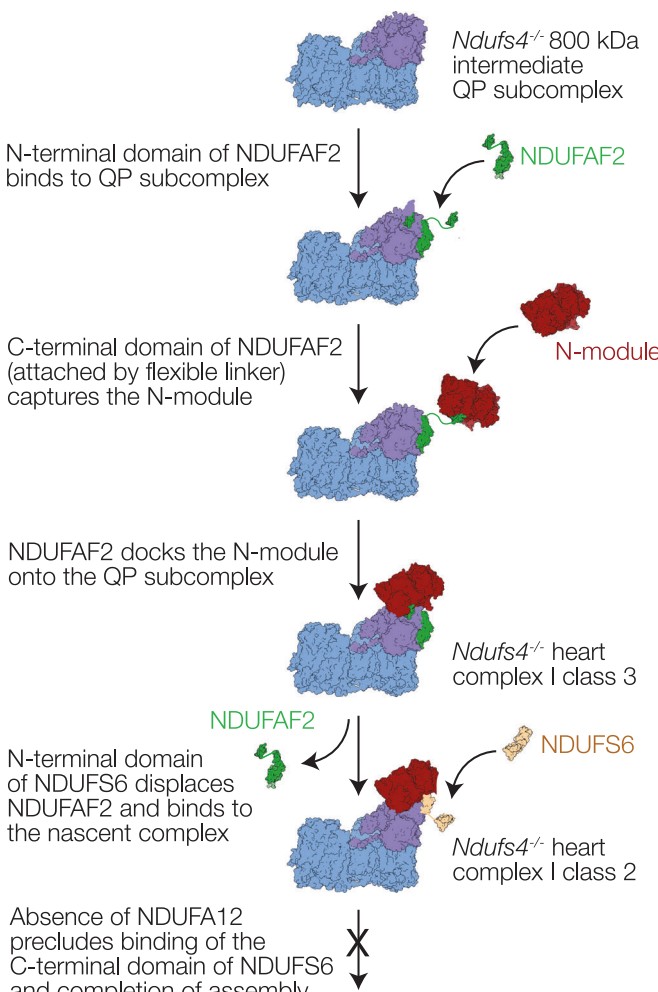

**Figure 7. Schematic diagram showing our proposed assembly pathway for mouse *ndufs4⁻/⁻* complex I.**

NDUFAF2 contains two domains, connected by a flexible linker. NDUFS6 also contains two domains linked together; in the class 2 complex I structure only the N-terminal domain is bound and observed in the density map.

N-terminal domain of NDUFS6 but not NDUFAF2, is thus surprising: first, because it suggests that NDUFS6 alone is sufficient to dislodge NDUFAF2, and second, because NDUFA12 is absent despite the availability of its binding site. Whereas in *Y. lipolytica*, *ndufs4⁻/⁻* complex I binds both NDUFS6 and NDUFA12 to complete assembly without NDUFS4 (Parey et al, 2019), assembly of the mouse *ndufs4⁻/⁻* complex stalls as NDUFA12 and the C-terminal domain of NDUFS6 fail to associate. However, the difference in the molecular phenotypes can be assigned, not to structural effects, but to the near-complete absence of NDUFA12 in *ndufs4⁻/⁻* mouse tissues, embryonic fibroblasts, and NDUFS4-mutated Leigh syndrome patient cells (Adjobo-Hermans et al, 2020); similar observations were also made for *ndufs4⁻/⁻* mouse glutamatergic vestibular neurons, that are affected in Leigh syndrome (Gella et al, 2020). Why NDUFA12 is also absent, as well as NDUFS4, is currently unclear, as well as unfortunate, since it compounds the *ndufs4⁻/⁻* defect and its pathological consequences by preventing the final stages in assembly.

## Implications of the NDUFS4 mouse model for complex I-linked disease

The $ndufs4^{-/-}$ mouse displays a severe Leigh-like phenotype (Kruse et al, 2008), consistent with decreased complex I catalysis (which results in decreased ATP production) causing increased steady-state NADH and $O_2$ levels, which together may result in increased ROS production. The discovery that a hypoxic environment mitigates the $ndufs4^{-/-}$ mouse phenotype (Jain et al, 2016) suggested oxidative damage as a substantial contributor to disease, with emphasis later placed on locally increased $O_2$ in the $ndufs4^{-/-}$ mouse brain (Jain et al, 2019). These data are consistent with our assays on isolated $ndufs4^{-/-}$ mitochondrial membranes and complex I (Fig. 3), which support the concept that increased $ndufs4^{-/-}$ ROS production in mitochondria stems from the altered mitochondrial conditions (Kussmaul and Hirst, 2006), rather than from an increased molecular-level propensity of the $ndufs4^{-/-}$ complex I to produce ROS. However, this mechanism suggests that hypoxia should benefit all complex I-linked mitochondrial diseases that similarly compromise the rate of catalysis, not be specific to a particular subset (including the $ndufs4^{-/-}$ family) or a specific genetic background (or species). It is thus essential to determine how general an effect hypoxia-rescue is. Should it be specific to $ndufs4^{-/-}$ variants, then one possible explanation is increased susceptibility to $O_2$-induced FeS cluster degradation, based on the proximity of NDUFS4 to clusters along the chain (Fig. 1D). The cluster chain extends beyond the N-module, such that repair or replacement may require either complete regeneration of the complex, or disassembly and replacement of a larger proportion of the hydrophilic arm than proposed for either an N-module-specific salvage pathway (Szczepanowska et al, 2020), or enhanced turnover of subunits in the N-module and adjacent hydrophilic arm (Dieteren et al, 2012; Krishna et al, 2021). Differences between our $ndufs4^{-/-}$ and wild-type mouse complex I structures have also provided some intriguing hints for additional subtle molecular changes, including the presence of bound ubiquinone, changes to the active-deactive transition, and the substantially decreased catalytic activity, which we cannot currently explain through incomplete assembly of the enzyme, or instability of the N-module. Wider knowledge of the effects of hypoxia on different complex I defects with varying molecular phenotypes will thus be crucial for elucidating both the origins of individual defects, the mechanism of hypoxia-rescue, and the balance of enzyme synthesis, assembly and degradation that occurs through complex feedback loops in mitochondrial homeostasis.

# Methods

## Animals

All procedures were carried out in accordance with the UK Animals (Scientific Procedures) Act, 1986 (PPL: P6C97520A) and the University of Cambridge Animal Welfare Policy. Wild-type C57BL/6J mice were supplied by Charles River (United Kingdom). The $ndufs4^{+/-}$ mouse strain on a C57BL/6J background was provided by Professor Richard Palmiter (University of Washington) (Kruse et al, 2008). Male and female mice were maintained in a temperature- and humidity-controlled animal care facility with a 12-h light/12-h dark cycle and free access to water and food and sacrificed at ~4–5 weeks by cervical dislocation. To produce

constitutive $ndufs4$ knockout mice, mice heterozygous for the $ndufs4$ knockout ($ndufs4^{+/-}$) were bred together, and wild-type littermates used as controls (Kruse et al, 2008). The genotype of each mouse was confirmed by PCR.

## Preparation of mitochondrial membranes from mouse heart and kidney

Heart and kidney tissues were excised from $ndufs4^{-/-}$ and wild-type mice, and mitochondria isolated as described previously (Fernández-Vizarra et al, 2010). Heart tissues were immersed immediately in ice-cold buffer AT (10 mM Tris-HCl, 75 mM sucrose, 225 mM sorbitol, 1 mM ethylene glycol-bis (β-aminoethyl ether)-N, N,N′, N′-tetraacetic acid (EGTA) and 0.1% (w/v) fatty acid-free bovine serum albumin, pH 7.4 at 4 °C). Kidney tissues were stored in ice-cold buffer A (0.32 M sucrose, 1 mM ethylenediaminetetraacetic acid (EDTA), and 10 mM Tris–HCl, pH 7.4 at 4 °C).

The hearts were diced, washed, then homogenized in 10 mL buffer AT per gram of tissue by seven to ten strokes of a Potter–Elvehjem homogenizer fitted with a Teflon pestle at ~1,000 rpm. The homogenate was centrifuged (1000 × g, 5 min), then the supernatant was centrifuged (9000 × g, 10 min) to collect the crude mitochondria. The kidneys were diced, washed, then homogenized in 5 mL buffer A per gram of tissue by five to eight strokes of the homogenizer at ~600 rpm. The homogenate was centrifuged (1000 × g, 5 min), then the supernatant centrifuged (15,000 × g, 10 min) to collect the crude mitochondria. The pellets were suspended in resuspension buffer (20 mM Tris-HCl, 1 mM EDTA, 10% glycerol, pH 7.4 at 4 °C) to a protein concentration of ~10 mg mL⁻¹. Pierce protease inhibitor tablets were added (1 tablet to 50 mL suspension) and aliquots were frozen and stored at −80 °C.

Membranes were prepared as described previously (Agip et al, 2018), where mitochondrial suspensions were thawed on ice, sonicated using a Q700 Sonicator (Qsonica, USA), at 65% amplitude with three 5-s bursts of sonication interspersed by 30-s intervals on ice and centrifuged at 75,000 × g for 1 h. The pellets containing the mitochondrial membranes were homogenized in resuspension buffer to ca. 5 mg mL⁻¹ and stored at −80 °C.

## Blue-native (BN) PAGE

For BN PAGE (Schägger and von Jagow, 1991), mitochondrial membranes were solubilized using 1% dodecyl-β-D-maltoside (DDM) (Glycon, Germany) for 30 min, centrifuged to remove insoluble materials (32,000 × g, 30 min), and run on Native PAGE™ Novex 3–12% Bis-Tris gels (ThermoFisher Scientific) according to the manufacturer's instructions, except that the gel was run for 30 min. at 100 V, then the inner buffer was exchanged for one containing 1/10th of the standard cathode buffer and run for a further 1.5 h at 180 V. Proteins were visualized using colloidal Coomassie R250, or the gels de-stained with MilliQ water for in-gel complex I activity assays using 100 μM NADH and 1 mg mL⁻¹ nitroblue tetrazolium (NBT) (Wittig et al, 2007).

## Catalytic activity measurements

Assays on membranes were carried out in 10 mM Tris-SO₄ (pH 7.4) and 250 mM sucrose at 32 °C in a SPECTRAmax

PLUS384 spectrophotometer (Molecular Devices, UK) using 10 µg-protein mL$^{-1}$ membranes supplemented by 1.5 µM horse heart cytochrome *c* and 15 µg mL$^{-1}$ alamethicin. NADH:O$_2$ oxidoreduction was initiated by the addition of 200 µM NADH, monitored at 340–380 nm ($\varepsilon = 4.81$ mM$^{-1}$ cm$^{-1}$) and its inhibitor sensitivity checked by the addition of 1 µM piericidin A (Sigma). Succinate:O$_2$ oxidoreduction was initiated by addition of 5 mM succinate, monitored using the reduction of 2 mM NADP$^+$ at 340–380 nm ($\varepsilon = 4.81$ mM$^{-1}$ cm$^{-1}$) by a coupled assay system comprising 90 µg mL$^{-1}$ fumarate hydratase (FumC) and 500 µg mL$^{-1}$ oxaloacetate decarboxylating malic dehydrogenase (MaeB) with 2 mM MgSO$_4$ and 2 mM K$_2$SO$_4$ (Jones and Hirst, 2013), and its inhibitor sensitivity checked by addition of 2 µM atpenin A5 (Santa Cruz Biotechnology, USA). Flavin-site activity measurements were carried out in the presence of 15 µg mL$^{-1}$ alamethicin and 2 µM piericidin A as follows: NADH:FeCN oxidoreduction used standard concentrations of 200 µM NADH and 1 mM FeCN and was monitored using $\varepsilon = 4.81$ mM$^{-1}$ cm$^{-1}$ at 340–380; NADH:APAD$^+$ oxidoreduction used 200 µM NADH and 1 mM APAD$^+$ ($\varepsilon = 3.16$ mM$^{-1}$ cm$^{-1}$ 450–500 nm) and NADH:HAR oxidoreduction used 100 µM NADH and 3.5 mM HAR ($\varepsilon = 4.81$ mM$^{-1}$ cm$^{-1}$ at 340–380 nm).

H$_2$O$_2$ production was measured using 10 U mL$^{-1}$ horseradish peroxidase, 10 µM Amplex red and 30 µM NADH with the background rates in the presence of catalase (5000 Units mL$^{-1}$) and in the absence of protein subtracted (Kussmaul and Hirst, 2006). Assays using purified complex I were carried out similarly at 32 °C in 20 mM Tris-Cl pH 7.5 (in the absence of alamethicin, cytochrome *c* or piericidin A). NADH:DQ oxidoreduction was measured using 100 µM DQ and 100 µM NADH in the presence of 0.15% soy bean asolectin and 0.15% 3-[(3-cholamidopropyl) dimethylammonio]-1-propanesulfonate (CHAPS).

## Preparation of complex I from mitochondrial membranes

*Ndufs4*$^{-/-}$ complex I was prepared as described previously with minor alterations (Agip et al, 2018; Yin et al, 2021). 3–4 mL of membrane suspension were solubilized by addition of 1% DDM, along with 0.005% phenylmethane sulfonyl fluoride (PMSF, Sigma-Aldrich, UK) for 30 min. 0.25 mM of the bis(sulfosuccinimidyl) suberate) (BS$^3$) cross-linker was added 5 min. after the start of the solubilization, and the cross-linking reaction quenched after 25 min. (at the end of the solubilization) by the addition of 50 mM Tris-HCl from a 1 M stock solution (pH 7.5, 4 °C). Following centrifugation (32,000 × *g*, 30 min), the solubilized enzymes were loaded onto a 1 mL Q-sepharose HP column (GE Healthcare, UK) pre-equilibrated in buffer containing 20 mM Tris-HCl (pH 7.14 at 20 °C), 1 mM EDTA, 0.1% DDM, 10% (v/v) ethylene glycol, 0.005% asolectin (Avanti Polar Lipid, USA) and 0.005% CHAPS (Calbiochem, Merck, Germany), and eluted by an increasing proportion of the same buffer plus 1 M NaCl. Complex I eluted in ~35% buffer B. The complex I-containing fractions were collected, concentrated using a 100 kDa MWCO Vivaspin 500 concentrator (Sartorius, Germany), and eluted from a Superose 6 Increase 5/150GL size exclusion column (GE Healthcare, UK) in 20 mM Tris-HCl (pH 7.14 at 20 °C), 150 mM NaCl and 0.05% DDM. Protein concentrations were determined using the BCA assay (Fisher Scientific UK). For the heart samples for cryo-EM analyses, 2.5 mM BS$^3$ was added for 30 min. on ice immediately before cryo-EM grid preparation.

## Cryo-EM data collection

UltrAuFoil gold grids (R0.6/1, Quantifoil Micro Tools GmbH, Germany) were prepared as described previously (Blaza et al, 2018). Briefly, they were glow discharged (20 mA, 90 s), incubated in a solution of 5 mM 11-mercaptoundecyl hexaethyleneglycol (SensoPath Technologies, USA) in ethanol (Meyerson et al, 2014) for 48 h in an anaerobic glovebox, then washed with ethanol and air-dried just before use. Using an FEI Vitrobot Mark IV, 2.5 µL of complex I solution (2.55 mg mL$^{-1}$ for kidney CI, 3.82 mg mL$^{-1}$ for heart CI) was applied to each grid at 4 °C in 100% humidity and blotted for 9–10 s at blotting force setting -10, before the grid was frozen by plunging it into liquid ethane. The highest quality cryo-frozen grids were identified using a 200 keV Talos Arctica microscope. Datasets for reconstruction were then collected using a Titan Krios (300 keV) microscope at the UK National Electron Bio-Imaging Centre at the Diamond Light Source.

A total of 3373 micrographs from the *ndufs4*$^{-/-}$ mouse kidney complex I sample were collected with a Falcon III detector in integrating mode, each with 59 movie frames, using the FEI EPU software. The calibrated pixel size was 1.46 Å, the defocus range was −2.1 to −3.3 µm and the total electron dose was 49.24 electrons Å$^{-2}$ s$^{-1}$ over a total exposure time of 1.5 s for each micrograph. Following inspection, 1982 micrographs were retained for analysis. A total of 7309 micrographs from the *ndufs4*$^{-/-}$ mouse heart complex I sample were collected with a Gatan K3 camera in super-resolution mode, each with 25 movie frames, using the FEI EPU software with aberration-free image shift. The calibrated pixel size was 1.352 Å, the defocus range was −1.5 to −2.9 µm and the total electron dose was 45 electrons Å$^{-2}$ s$^{-1}$ over a total exposure time of 5.4 s for each micrograph.

## Cryo-EM data processing for complex I from *ndufs4*$^{-/-}$ mouse kidney

Cryo-EM data processing was carried out using RELION 3.0 (Scheres, 2012). Micrographs were motion corrected using Motioncor2 with dose-weighting (Zheng et al, 2017) followed by CTF estimation using Gctf (Zhang, 2016). 149,430 particles were automatically picked and extracted with a box size of 360 pixels. The particles were imported to cryoSPARC 3.3.2 (Punjani et al, 2017) and first 2D classified to remove non-protein junk. The resulting 115,461 particles went through rounds of ab initio and heterogeneous refinements to produce 3D volumes that were used to re-classify them into 6 classes. The class resembling intact complex I contained 19,260 particles and was further subclassified through ab initio and heterogeneous refinements to remove particles that lacked the N-module of the complex. This approach yielded 7563 particles that were refined using non-uniform refinement, minimizing per-particle scales, and optimizing defocus, beam tilt, and trefoil. The final mask for post processing was created in RELION using a map generated from the model of the *ndufs4*$^{-/-}$ mouse heart class 3 using the molmap command in Chimera (Pettersen et al, 2004) with a binary extension of 4 pixels and a soft edge of 8 pixels. Postprocessing was performed on the final map using RELION. The global resolution of the map was 6.2 Å based on the FSC = 0.143 criterion.

## Cryo-EM data processing for complex I from *ndufs4*[−/−] mouse heart

Micrographs were motion corrected using Motioncor2 in RELION 3.1 and imported into cryoSPARC v3 (Punjani et al, 2017) followed by CTF estimation using Gctf (Zhang, 2016). The initial auto-picking process obtained 1,141,407 particles, which were cleaned by two rounds of 2D classifications resulting in 84,141 complex I-like particles. The particles were further cleaned by a reference-free ab initio 3D classification and 71,761 were selected. These particles were exported into RELION 3.1 (Zivanov et al, 2020) and subjected to CTF refinement and Bayesian polishing using all frames. The polished particles were exported to cryoSPARC and subjected to rounds of 3D classification; first the particles were homogeneously refined, and then classified into two classes using a very generous mask including the entire micelle, and a target resolution of 20 Å. The two classes were then each subclassified; one subclass contained additional density to the side of the micelle, and another contained density in the region where NDUFA12 is usually found. Both classes were subclassified into two classes using a mask around the areas of interest and target resolutions of 10 and 15 Å, respectively. Two of the resulting maps were found to be essentially identical and were pooled together, and all three maps were refined using non-uniform refinement. The two maps with additional densities still suffered poorer clarity of these areas than the bulk of the protein, and so they were subclassified once more using masks around the respective additional density areas and using the refined volumes (with and without the additional density) as input models. This resulted in a further class that appeared essentially identical to the class without additional density, and so all these particles were pooled together. A class of 3035 particles that appeared incompletely classified in the region of additional density were excluded from further analyses. The three classes identified were refined by non-uniform refinement using per particle scale factors, and further postprocessed in RELION using a mask generated with a soft edge of 8 pixels, from a volume made in UCSF Chimera (Pettersen et al, 2004) from a preliminary PDB model based on 6ZR2 (Bridges et al, 2020), with the addition of AlphaFold models AF-P50544 for ACADVL and AF-Q59J78 for NDUFAF2 (Jumper et al, 2021) as appropriate. Global resolutions are reported at FSC = 0.143. Particle subtraction was performed in cryoSPARC on the ACAD-containing map in the region of the ACAD dimer using a mask generated with a soft edge of 8 pixels based on a crudely placed volume made from a dimer of ACADVL in UCSF Chimera. Subtracted particles were reconstructed, and the volume aligned to find the symmetry axis, for local refinement with C2 symmetry enforced.

## Model building and refinement for complex I from *ndufs4*[−/−] mouse heart

Model building and refinement were performed using Coot 0.9.3 (Emsley et al, 2010) and Phenix-1.18.2-3874 (Afonine et al, 2018). The active mouse complex I (PDB: 6ZR2) with the deletion of NDUFS4 and NDUFA12 subunits was rigid body–fitted into the map using UCSF ChimeraX as the initial model. Where appropriate, AlphaFold models AF-P50544 and AF-Q59J78 (Jumper et al, 2021) were placed in the density using the fit-in-map function of ChimeraX, and fitted in initially using ISOLDE (Croll, 2018) in ChimeraX using default parameters. The models were then subject to cycles of manual adjustment in Coot 0.9.3 using the globally sharpened map and refined with secondary structure restraints by

phenix.realspace. The ubiquinone-9 molecule was manually fitted into the density and real space refined by Phenix. Model-to-map FSC curves were generated using the Phenix Comprehensive validation (cryo-EM) tools and final model statistics were produced by the Phenix implementations of MolProbity 4.4, and EMRinger.

## Complexome profiling experiments

Blue native PAGE/complexome profiling was performed as described previously (Bridges et al, 2017), based on the method developed by Heide et al. (Heide et al, 2012). The tryptic peptides from excised 1 mm slices (62 from *ndufs4*[−/−] and 64 from wild-type) from BN-PAGE gel lanes were analyzed by LC/MS/MS using nanoscale reverse-phase HPLC and an LTQ Orbitrap XL mass spectrometer (Thermo Scientific). Proteins were identified with the MASCOT algorithm (v2.4, Matrix Sciences) and the results from each gel slice compiled into a consensus report using Proteome Discoverer (Thermo Scientific). The MS data were compared with the Uniprot sequence database (August 2020 version containing 563,000 sequences) with peptide mass and fragment mass tolerances of 10 ppm and 0.5 Da, respectively. Variable protein modifications included methionine oxidation, cysteine propionamide, and protein N-terminal acetylation or formylation, with the potential for one missed tryptic cleavage site. Protein quantities in each gel slice were estimated from the average of the intensity values of the three highest intensity peptides.

## Protein identification

Proteins from 10 μg of purified complex I from *ndufs4*[−/−] mouse heart (not treated with cross-linker) were resolved by SDS PAGE. The gel was stained with Coomassie R-250, and blue stained bands were excised, digested with trypsin, and the peptides extracted as described previously (Shevchenko et al, 1996). Tryptic peptides were analyzed using either a MALDI-TOF-TOF mass spectrometer (Sciex Model 4800 Plus) using α-cyano-4-hydroxycinnamic acid as the matrix compound, or by LC-MS-MS in an Orbitrap XL mass spectrometer as described previously (Andrews et al, 2013). Peptide masses and peptide fragment data were identified as described above, except that the tolerances for peptide mass and fragment data were 70 ppm and 0.8 Da for MALDI-TOF-TOF data, or 5 ppm and 0.5 Da for Orbitrap data. Either 1 or 4 missed tryptic cleavages were permitted, respectively.

## Relative protein quantification

Tryptic digests were produced from 20 μg samples of mouse kidney mitochondrial membranes. Proteolysis was maximized by the presence of 0.5% (w/v) sodium deoxycholate in 100 mM ammonium bicarbonate and three additions of trypsin (1:100 w/w) over an incubation period of 48 h at 37 °C (Masuda et al, 2008). Deoxycholate was removed from the digest mixture by phase extraction with acidified ethyl acetate. Portions of the digests were analyzed using a Q-Exactive plus mass spectrometer (Thermo Scientific) as described previously (Bridges et al, 2017), and the data compared with the UNIPROT database comprising 17,107 mouse sequences as described above. The comparisons were performed with mass tolerances of 5 ppm and 0.01 Da for peptide masses and fragment ion masses, respectively, an allowance of two missed tryptic cleavages and the potential for methionine oxidation. These analyses were performed in triplicate.

Common peptides amongst the replicates from *ndufs4*$^{-/-}$ and wild-type samples were identified for each protein of interest (complexes I, II, III, IV, ACAD9 and ACAD-VL) and the VDAC1, VDAC2 and VDAC3 proteins used for normalization. The sum of the peptide areas for these common peptides were used to produce an abundance value for each protein or protein complex. The values were internally normalized for each technical replicate to the sum of common VDAC1 + VDAC2 + VDAC3 peptides and the average and standard deviation of the *ndufs4*$^{-/-}$ and wild-type samples were calculated for each protein or protein complex of interest. Relative abundance of proteins in the *ndufs4*$^{-/-}$ sample were calculated and expressed as a fraction of the wild-type value, with appropriate error propagation.

## Data availability

Cryo-EM maps and models have been deposited in the Electron Microscopy Data Bank (https://www.ebi.ac.uk/pdbe/emdb/) and Protein Data Bank (https://pdbj.org/) with accession numbers EMD-16398 and PDB:8C2S (*ndufs4*$^{-/-}$ heart class 1); EMD-16515 and PDB:8CA1 (class 1 ACADVL); EMD-16516 and PDB:8CA3 (*ndufs4*$^{-/-}$ heart class 2); EMD-16517 and PDB:8CA4 (*ndufs4*$^{-/-}$ heart class 2 N-module); EMD-16518 and PDB:8CA5 (*ndufs4*$^{-/-}$ heart class 3); and EMD-16514 (*ndufs4*$^{-/-}$ kidney). Raw cryo-EM images have been deposited into EMPIAR (https://www.ebi.ac.uk/pdbe/emdb/empiar/) with IDs: EMPIAR-11356 for complex I from *ndufs4*$^{-/-}$ mouse heart and EMPIAR-11606 for complex I from *ndufs4*$^{-/-}$ mouse kidney. The complexome datasets for mouse heart mitochondrial membranes have been deposited in the CEDAR database (https://www3.cmbi.umcn.nl/cedar/) with accession number CRX44, and the raw data for the unlabeled protein quantification from mouse kidney mitochondrial membranes have been deposited in the PRIDE database (https://www.ebi.ac.uk/pride/) with the accession code PXD044975.

## Peer review information

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

## Acknowledgements

This research was funded by the Medical Research Council (MC_UU_00015/2 and MC_UU_00028/1 to JH). We thank Shujing Ding, Ian M. Fearnley and Michael E. Harbour (MBU proteomics facility) for mass spectrometry analyses, James N. Blaza (MBU, now University of York) for assistance with preliminary cryo-EM experiments, Carlo Viscomi for assistance with mouse husbandry, Diamond for access and support of the cryo-EM facilities at the UK national electron Bio-Imaging Centre (eBIC), proposals EM17057-2 and BI22238-4, funded by the Wellcome Trust, MRC, and BBSRC, as well as staff at University of Cambridge cryo-EM facility for grid screening.

## Author contributions

**Zhan Yin**: Data curation; Formal analysis; Investigation; Methodology; Writing—original draft; Writing—review and editing. **Ahmed-Noor A Agip**: Formal analysis; Investigation; Methodology; Writing—review and editing. **Hannah R Bridges**: Data curation; Formal analysis; Supervision; Validation; Investigation; Visualization; Methodology; Writing—original draft; Writing—review and editing. **Judy Hirst**: Conceptualization; Supervision; Funding acquisition; Methodology; Project administration; Writing—review and editing.

## Disclosure and competing interests statement

The authors declare no competing interests.

# Expanded View Figures

**Figure EV1.   Complexome profiles from BN-PAGE analyses of solubilized mitochondrial membranes from (A) wild-type and (B) *ndufs4⁻ᐟ⁻* mouse heart.**

The heatmap represents the relative protein abundance in each gel slice (based on the peak areas of the three top-scoring peptides for each protein from LC-MS analyses) with the data for each protein normalized to the highest intensity signal within the lane. The prefix "NDU" has been omitted from subunit labels for brevity. Where a protein was not detected in one sample, the column is completely black. Subunits ND3, ND4L and ND6 were not detected in either sample. The masses of complex I-related bands specific to *ndufs4⁻ᐟ⁻* membranes of ~800 (QP subcomplex) and ~200 kDa (N module) were estimated using the migration and masses of known complexes (inset). Source data are available online for this figure.

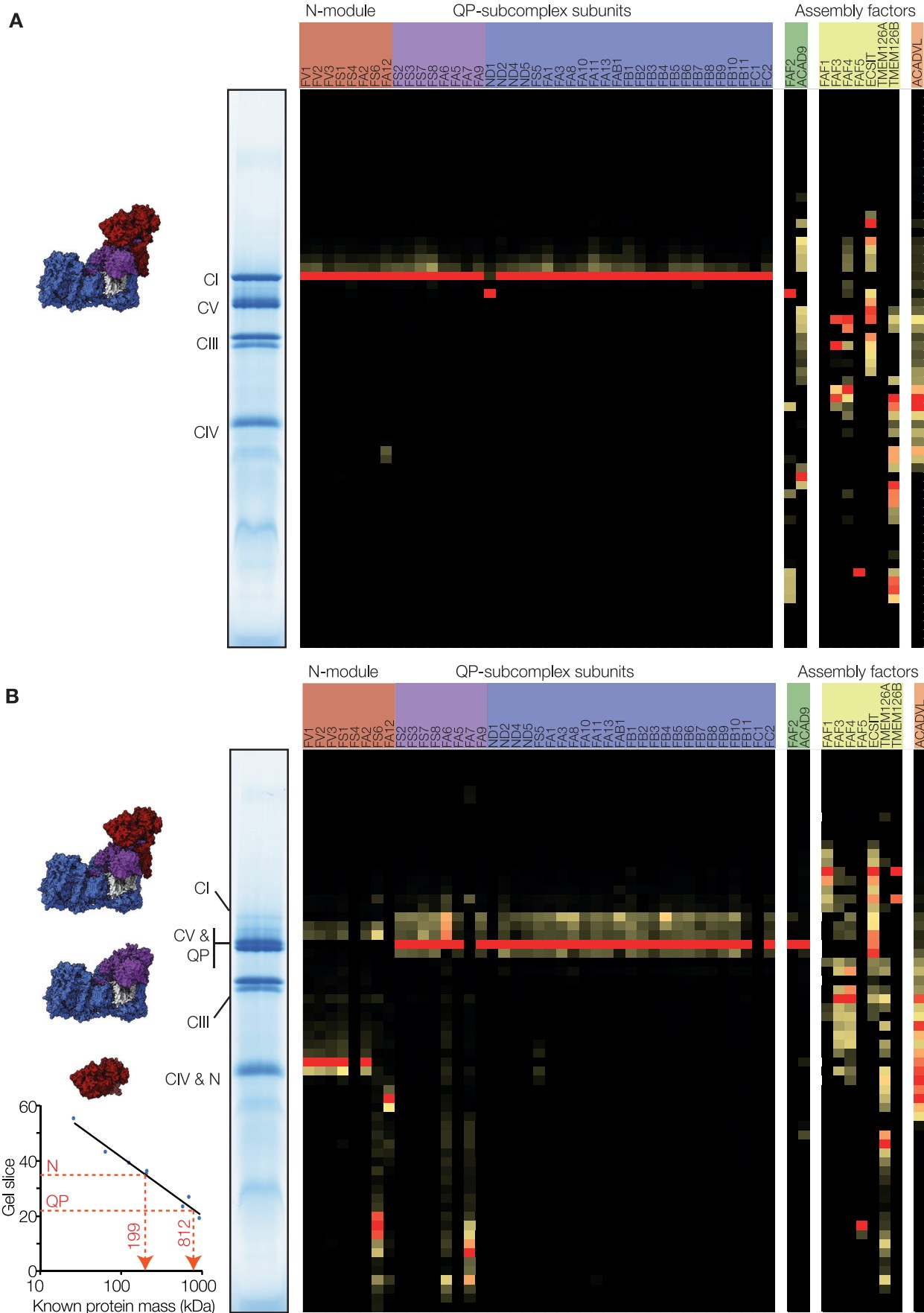

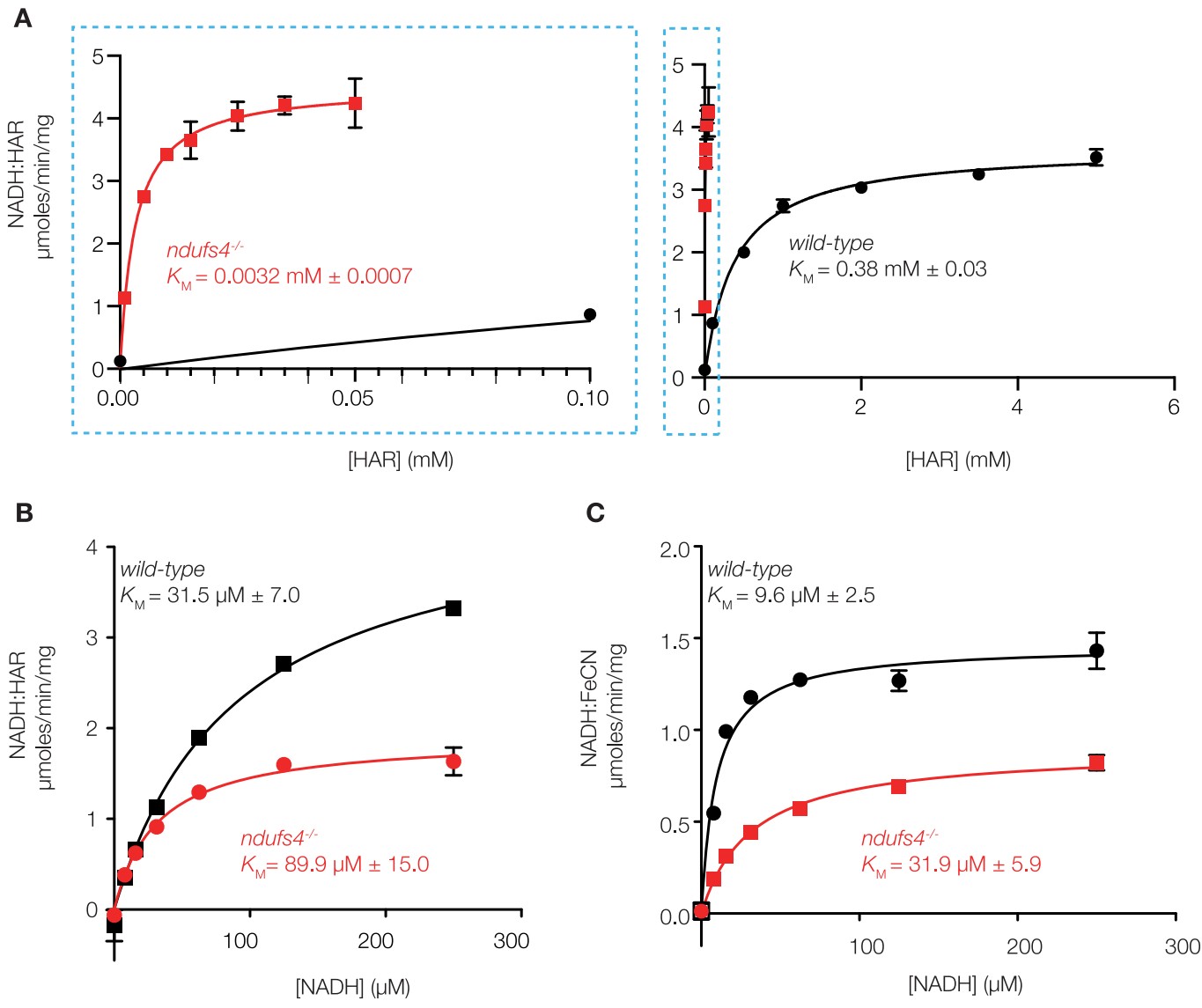

**Figure EV2. Flavin-site kinetics in *ndufs4*$^{-/-}$ mitochondrial membranes compared to wild-type.**

(**A**) Kinetic evaluation of the Michaelis constant ($K_M$) for HAR in the NADH:HAR oxidoreductase reaction catalyzed by mouse heart membranes in the presence of 100 µM NADH. (**B**) Kinetic evaluation of the Michaelis constant ($K_M$) for NADH in the presence of 1 mM HAR. (**C**) Kinetic evaluation of the Michaelis constant ($K_M$) for NADH in the NADH:FeCN oxidoreductase reaction catalyzed by mouse kidney membranes in the presence of 100 µM FeCN. Due to the scarcity of materials, the assays in panels **A** and **B** were performed on different membrane samples, so the absolute rates are not directly comparable between them. Data points are mean averages ± SEM (n = 3 technical replicates). Source data are available online for this figure.

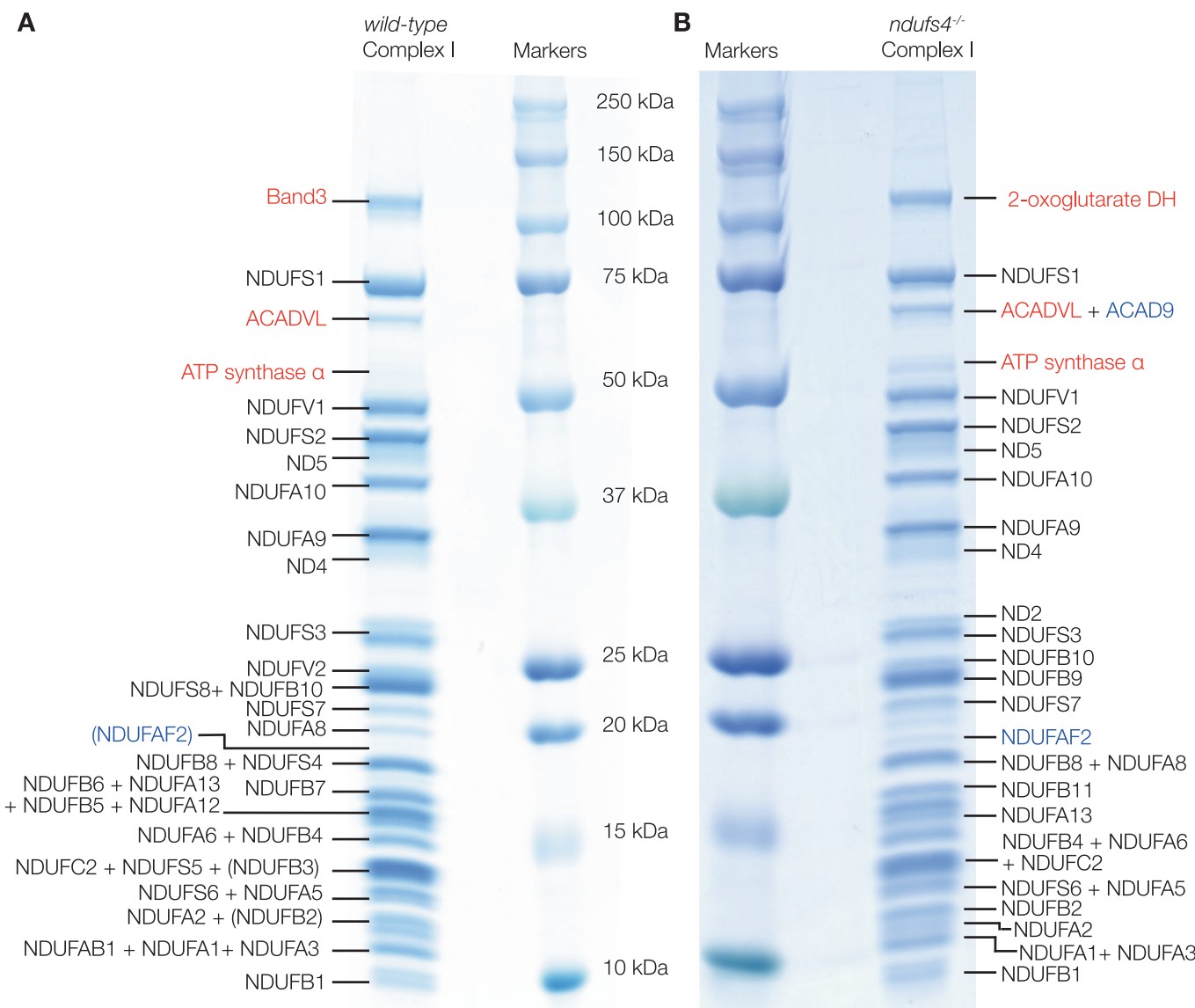

**Figure EV3. SDS PAGE of complex I purified from (A) wild-type and (B) *ndufs4*<sup></sup>−/− heart mitochondrial membranes.**

Bands were manually excised, and proteins identified by MALDI-TOF-TOF analyses of tryptic digests. Complex I subunits are labeled in black, known assembly factors in blue, and other proteins detected in red. For proteins labeled in brackets individual peptide scores were below the 95% threshold but their protein scores were above the peptide threshold. Information on protein identification is in Appendix Tables S1 and S2.

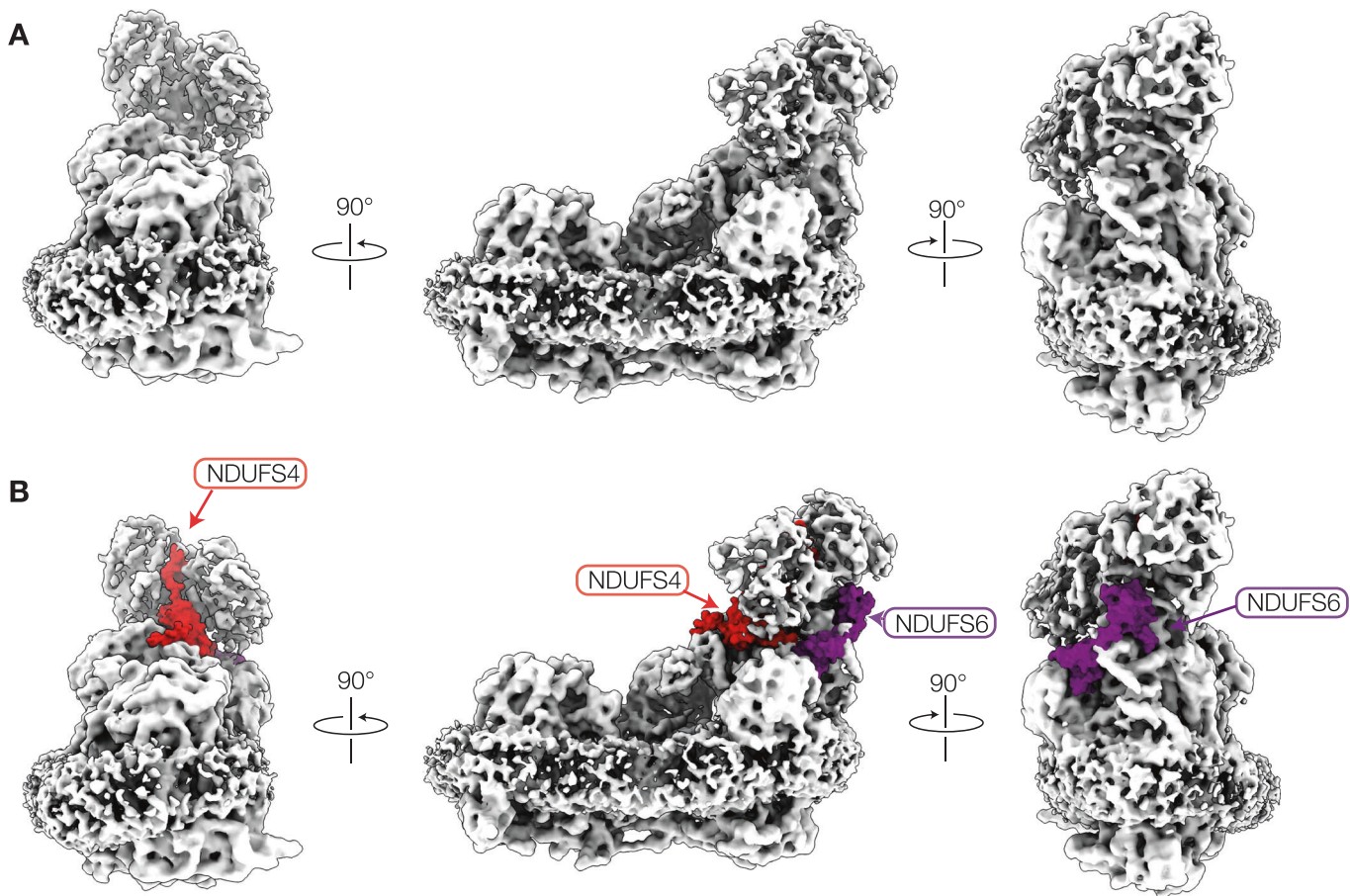

**Figure EV4.  Views of the cryo-EM density map of complex I from *ndufs4*$^{-/-}$ kidney and comparison to the model for wild-type heart complex I.**

(**A**) The density map of complex I from *ndufs4*$^{-/-}$ kidney. (**B**) The wild-type complex I model (PDB:6ZR2) is docked into the *ndufs4*$^{-/-}$ complex I density map (gray, EMD:16514); no density is present in the locations of NDUFS4 (red model) and NDUFS6 (purple model).

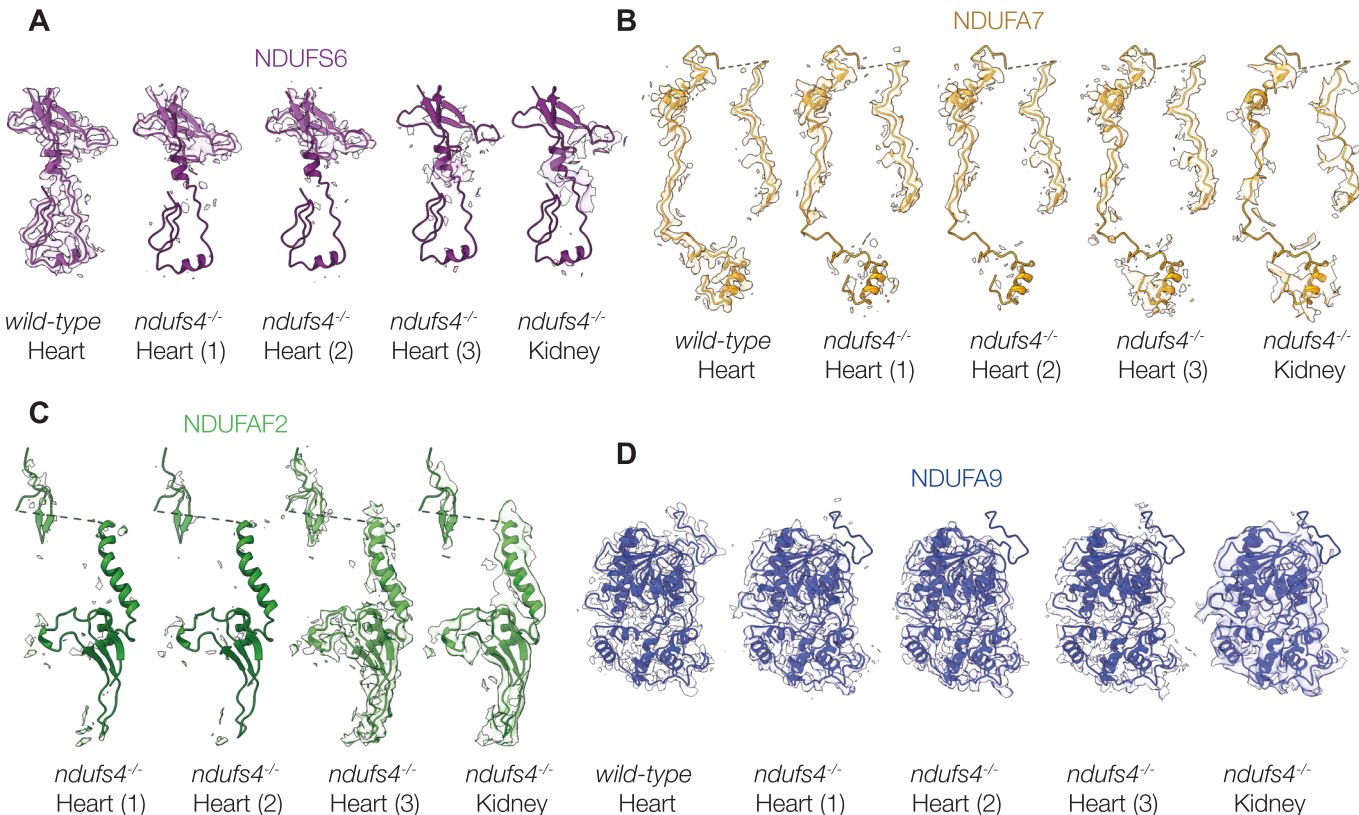

**Figure EV5. The cryo-EM densities for NDUFS6, NDUFA7, NDUFAF2 and NDUFA9.**

The models for wild-type heart complex I (PDB: 6ZR2, for NDUFS6, NDUFA7 and NDUFA9) and the class 3 model (for NDUFAF2) from were fitted into each density map using the fit-in-map function in Chimera X. The density within 3 Å of each fitted model is shown for (A) subunit NDUFS6; (B) subunit NDUFA7; (C) assembly factor NDUFAF2; and (D) subunit NDUFA9. They are shown for the map of wild-type heart complex I (EMD-11377); the *ndufs4$^{-/-}$* heart complex I class 1, 2 and 3 maps; and the *ndufs4$^{-/-}$* kidney complex I map, with map thresholds of 0.027, 1.76, 2.18, 1.52 and 1.12, respectively.

