## [Peer Review File · The EMBO Journal]

Structural insights into respiratory complex I deficiency and assembly from the mitochondrial disease-related *ndufs4*^{-/-} mouse

Zhan Yin, Ahmed-Noor Agip, Hannah Bridges, and Judy Hirst
DOI: [10.15252/emboj.2023114973](https://doi.org/10.15252/emboj.2023114973)

Corresponding author(s): Judy Hirst (jh480@cam.ac.uk) , Hannah Bridges (hrb@mrc-mbu.cam.ac.uk)

Review Timeline:

Submission Date:	15th Jul 23
Editorial Decision:	19th Aug 23
Revision Received:	6th Sep 23
Editorial Decision:	24th Oct 23
Revision Received:	30th Oct 23
Accepted:	7th Nov 23

Editor: *Cornelius Schneider*

Transaction Report:

Dear Dr. Hirst,

Thank you for submitting your manuscript for consideration by the EMBO Journal.
We have now received comments from three reviewers, which are included below for your information.

As you will see from the reports, the reviewers appreciate the work, while also indicating a number of constructive points that would need to be addressed before acceptance here. From my side, I find the reviewer comments reasonable and constructive. Therefore, based on these positive assessments, I would like to invite you to address the issues raised by the reviewers in a revised manuscript. I would be happy to discuss the revision in more detail via email or phone/videoconferencing.

We generally allow three months as standard revision time. As a matter of policy, competing manuscripts published during this period will not negatively impact on our assessment of the conceptual advance presented by your study. However, please contact me as soon as possible upon publication of any related work to discuss the appropriate course of action. Should you foresee a problem in meeting this three-month deadline, please contact us to arrange an extension.

When preparing your letter of response to the referees' comments, please bear in mind that this will form part of the Review Process File and will therefore be available online to the community. For more details on our Transparent Editorial Process, please visit our website: <https://www.embopress.org/page/journal/14602075/authorguide#transparentprocess>. Please also see the attached instructions for further guidelines on preparation of the revised manuscript.

Please feel free to contact me if you have any further questions regarding the revision. Thank you for the opportunity to consider your work for publication, and I look forward to your revision.

With best regards,

Cornelius Schneider

Yours sincerely,

Cornelius Schneider, PhD
Editor
The EMBO Journal
c.schneider@embojournal.org

We realize that it is difficult to revise to a specific deadline. In the interest of protecting the conceptual advance provided by the work, we recommend a revision within 3 months (17th Nov 2023). Please discuss the revision progress ahead of this time with the editor if you require more time to complete the revisions. Use the link below to submit your revision:

Referee #1:

In this manuscript by the renowned group of Judy Hirst, cryo-EM analysis is applied to investigate the consequences of NDUFS4 absence in kidney/heart samples from the *Ndufs4*^{-/-} mouse model. Its main findings are that NDUFS4 absence induces:

- 1) Loose association of the NADH-dehydrogenase (N) module
- 2) Formation of a class containing NDUF2F2 (substituting NDUF2A12)
- 3) Formation of a class containing NDUFS6
- 4) Absence of NDUF2A12

It is proposed that NDUF2F2 is a recruiter of the N-module during CI assembly. The manuscript is easy to read, well illustrated, and experiments/figures are presented clearly.

However, in the literature there is already quite some information available on the *Ndufs4* subunit in relation to human disease and CI biogenesis/stability. In this sense, virtually all of the relevant studies are properly cited in the current manuscript and (indirect) evidence for findings 1, 2 and 4 is already presented elsewhere.

MAJOR COMMENTS:

line 200-201: how does this statement relate to a previous study concluding that *ndufs4* mutations in patient cells primarily lower CI levels and not intrinsic activity (PMID: 20818732)?

line 205-207/214-216: this idea is not new. Please include references.

line 228-229: how does the reduced rate of H₂O₂ generation relate to the increased ROS levels observed in patient cells with *Ndufs4* mutations?

line 425-426: it was demonstrated in the literature that MEFs derived from the *Ndufs4*^{-/-} mouse display rotenone-sensitive oxygen consumption, meaning that CI is active in situ (PMID: 22430089).

line 530/531 and 535-536: how do the authors explain the presence of NDUF2F2 in active CI on BN-PAGE in patient fibroblasts (PMID: 32335026)? This strongly argues that NDUF2F2 acts as a stabilizing bridge.

MINOR COMMENTS:

line 35-36: Does CI also contribute to ROS generation during non-pathological conditions?

line 42-44: Please state that the NDUFAB1 subunit occurs twice in the mature complex.

line 102: do the authors mean that both intact and 830-kDa subcomplex are observed for the same sample on the same BN-PAGE gel? Please explain.

line 570: how can decreased CI catalysis cause increased steady-state oxygen levels?

Referee #2:

The manuscript by Yin and colleagues reports cryoEM structures of complex I/complex I assembly intermediates isolated from a *ndufs4* knock-out mouse line. In humans, mutations in *NDUFS4* are a frequent cause of Leigh syndrome.

The migration behavior of complex I and assembly factors in mitochondrial membranes was carefully analyzed by complexome profiling. The most prominent species was the QP complex lacking the N module. A complex I variant with all modules was a minor species. Assembly factor *NDUFAF2* comigrated with the QP subcomplex. It is surprising that also *ACAD9* appears to be associated with this subcomplex. *ACAD9* is required for beta oxidation but is also a central component of the MCIA complex, together with *ECSIT* and *TMEM126B*. The latter two subunits are mainly found in the region of supercomplexes and separated from *ACAD9*. A comment on this unexpected behavior would be welcome. It may also help to give a bit more background on *ACAD9* and *ACADVL* in the manuscript. Please also note that *ACAD9* is deflavinated when bound to *ECSIT*.

Electron transfer activity with different acceptors and ROS formation were determined. The authors did not detect an increased formation of ROS. However, I miss a comment here on how specific their assay is. The Amplex red test monitors H_2O_2 but would it also detect superoxide? This question seems important to me to understand the pathogenesis of *NDUFS4* dysfunction and why exposure to low oxygen concentrations is beneficial. Are there alternatives for the Amplex red assay?

For structure determination complex I was purified by chromatography in detergent. Cross-linking stabilized the fragile complex(es) and permitted to determine structures at up to 2.9 Å resolution. Given the small sample size, this is impressive. However, cross-linking may have also caused tight binding of the *ACADVL* close to the "heel" of the L shaped particle (class 1) which is discussed as a possible artefact. In line with a previous publication, *NDUFA12* was not detected in any of the three classes and the authors propose that mainly *NDUFS6* is responsible for detachment of the *NDUFAF2* assembly factor in the mutant (Fig. 7). It is discussed that in the *NDUFS4* knock-out, *NDUFA12* is greatly diminished (ref Adjobo-Hermans et al.). However, according to the complexome profiling analysis in this study, this seems not to be the case (or normalization of signal strength in the heat map needs more explanation). *NDUFA12* is present and migrates as a lower molecular weight species. Please comment. And it would help to somehow improve the labelling in Fig. EV1.

Taken together, this is a very well written manuscript that offers new insight into mitochondrial disease and complex I assembly based on excellent cryoEM data.

Referee #3:

In this manuscript, Yin et al describe the Cryo-EM structure of the complex I species present in mouse mitochondria that lack the subunit *Ndufs4*. This is particularly relevant, as this genetic mouse model is one of the few animal models that is available for studying complex I deficiency (particularly as a model of Leigh Syndrome) and has been used extensively in the field of mitochondrial disease and complex I biology. It is established that in the absence of the subunit *NDUFS4*, the catalytic N-module of complex I is destabilised from the rest of the complex by blue native PAGE analysis (Calvaruso et al 2012; Stroud et al 2016), yet there is still residual complex I activity detected in these mitochondria. Here, the authors address this by using chemical crosslinking on isolated mitochondria and demonstrate that in vivo, the N module remains mostly attached and that this crosslinked complex I can be used for further structural determination. The structural studies identified several classes of complex I intermediates in the absence of *NDUFS4*, notably including a structure with the complex I assembly factor *NDUFAF2*.

Overall, this manuscript provides structural detail for an important mitochondrial disease-causing mutation, however, this reviewer feels that many statements made are over-interpreted in the context of the data presented and the absence of further biochemical data. In order to be considered for publication in EMBO Journal, the authors should address the following points
Major points:

1. The results shown in figure EV1 should include the subunit/assembly factors as part of the figure to identify each protein being described easily.
2. The structure of *Ndufaf2* in the Heart class 3 is very interesting (Figure 5B). Given that a substantial amount of the protein density is present in the CryoEM structure, can the authors use the alphafold model for *Ndufaf2* to predict the missing density?
3. The discussion surrounding the presence of the ACAD proteins present in the Cryo-EM structure is quite confusing and convoluted for a result that is artefactual. The authors should consider revising this section to improve clarity to a broader audience.
4. In the discussion, the authors state that the binding of *NDUFS4*, *NDUSF6* and *NDUFA12* is required to displace *NDUFAF2*

(line 540). In two studies using NDUFA12 patient-derived cells or modified human cells lacking NDUFA12, it appears that complex I can assemble, is active and has similar migration to wildtype complex I and stably bind to NDUFAF2 (Adjobo-Hermans et al 2020; Stroud et al 2016). To me, this would suggest that NDUFAF2 displacement predominantly requires NDUFA12 recruitment, as NDUFAF2 can still bind to complex I in the presence of NDUFS4 and NDUFS6. The authors should include this in the context of their structural data.

5. The role of NDUFAF2 is elusive in complex I assembly, though patients with homozygous null mutations in this protein can still assemble complex I (to a lesser extent than control cells; Ogilvie et al 2005). While NDUFAF2 might participate in N-module recruitment, it cannot be excluded that other proteins also participate in this process, given the clinical and biochemical evidence available. The authors should include this in their discussion.

Minor points:

1. It is unclear where the structures in Figure 1 come from- please cite the appropriate publication and PDB ID in text/figure legend.
2. Line 153-154: Four proteins were not detected by mass spectrometry, however, only three subunits were mentioned. What was the fourth subunit?
3. In Figures 3A and 3B is not clear what the absorbance at 420nm is measuring. This should be made clearer.
4. The authors should note
5. Line 305: 'NDUSF4' should be 'NDUFS4'
6. Line 525: At the start of this sentence, it is unclear what 'its' refers to.

MRC
Mitochondrial
Biology Unit

UNIVERSITY OF
CAMBRIDGE

06 September 2023

Dear Dr. Schneider,

Thank you for inviting us to revise our manuscript EMBOJ-2023-114973 entitled "Structural insights into complex I deficiency and assembly from the disease-related *ndufs4*^{-/-} mouse" for publication in *EMBO journal*. We were very pleased with the favourable assessments, and hope that you will find our responses to the review and editorial comments satisfactory, and our revised manuscript now suitable for publication.

Our responses to the comments from the reviewers are presented below and revisions are shown in red in the revised manuscript.

Please note that for Figure panels Fig. 2E, Fig. 3C & D and Fig. 6D, $n < 5$ but the data presented are values for *ndufs4*^{-/-} relative to *wild-type*, each measured with their own technical replicates, and therefore, plotting individual points on these graphs is not possible.

Thank you for your consideration and we look forward to your positive response.

Yours sincerely,

Dr. Hannah Bridges

Investigator Scientist

T: +44 (0)1223 252812

E: hrb@mrc-mbu.cam.ac.uk

Prof Judy Hirst FRS FMedSci

Professor and Unit Director

T (PA): +44 (0)1223 252704

E: jh@mrc-mbu.cam.ac.uk

Medical Research Council Mitochondrial Biology Unit

University of Cambridge, The Keith Peters Building, Cambridge Biomedical Campus, Hills Road,
Cambridge CB2 0XY

T: +44 (0)1223 252700 mrc-mbu.cam.ac.uk [@MRC_MBU](https://twitter.com/MRC_MBU)

Reviewer 1

In this manuscript by the renowned group of Judy Hirst, cryo-EM analysis is applied to investigate the consequences of NDUFS4 absence in kidney/heart samples from the *Ndufs4*^{-/-} mouse model. Its main findings are that NDUFS4 absence induces:

- 1) Loose association of the NADH-dehydrogenase (N) module
- 2) Formation of a class containing NDUFAF2 (substituting NDUFA12)
- 3) Formation of a class containing NDUFS6
- 4) Absence of NDUFA12

It is proposed that NDUFAF2 is a recruiter of the N-module during CI assembly. The manuscript is easy to read, well illustrated, and experiments/figures are presented clearly.

We thank the reviewer for their summary and positive comments.

However, in the literature there is already quite some information available on the *Ndufs4* subunit in relation to human disease and CI biogenesis/stability. In this sense, virtually all of the relevant studies are properly cited in the current manuscript and (indirect) evidence for findings 1, 2 and 4 is already presented elsewhere.

We understand the reviewer's point that indirect evidence for three of our main conclusions is already available in the literature. As noted by the reviewer we have clearly cited and described these previous studies. Our study now provides direct structural evidence to complement these previous observations, which were all made using different techniques, and draws them together to provide added value and a new understanding under a common molecular-level framework.

MAJOR COMMENTS:

line 200-201: how does this statement relate to a previous study concluding that *ndufs4* mutations in patient cells primarily lower CI levels and not intrinsic activity (PMID: 20818732)?

We agree that the content of the enzyme is consistently lower in the *ndufs4* knockout across numerous studies, but as we have discussed in our manuscript quantification of the amount of intact enzyme present is very challenging (and is required to determine the intrinsic activity). We note that a more recent study from the same group (PMID: 32335026) found a 50% reduction in complex I subunits in mouse tissues (consistent with our results) and also determined 7-14% of the WT enzymatic activity in *ndufs4* patient fibroblasts (also consistent with our results), indicating that there is less consistent agreement on the intrinsic activity. However, in considering this comment we have now realised that we should acknowledge the possibility of our enzyme activity being decreased by our preparation, as has been noted for preparations involving the *ndufs4*-knockout enzyme in other contexts. Therefore, we have now included the following text (lines 203-210 of the revised manuscript):

"Our data thus suggest that in our samples, the lower NADH:O₂ activity in kidney membranes is due to both decreased content and decreased intrinsic catalytic activity of the *ndufs4*^{-/-} enzyme, which

we estimate to catalyze at ~40% of the wild-type rate. We note that previous studies have observed disparities in the activity of complex I in intact cells or tissues compared to isolated mitochondria (PMID: 18396137, PMID: 22430089), and so cannot discount the possibility that, due to our preparatory procedure, the activity we observe in membranes is an underestimation of the activity *in vivo*.”

line 205-207/214-216: this idea is not new. Please include references.

We apologise for this omission; we have now amended the text at lines 213-216 in the revised manuscript as follows:

“This observation suggests that, as suggested previously (PMID: 32335026), the N-module can remain attached to the QP subcomplex *in vivo* and in membrane preparations but readily dissociates during BN-PAGE.”

line 228-229: how does the reduced rate of H₂O₂ generation relate to the increased ROS levels observed in patient cells with *Ndufs4* mutations?

First, we note a degree of inconsistency in the literature about the level of ROS production in *NDUFS4* knockout cells (see lines 469-471). Second, in our work we have used isolated mitochondrial membranes and purified complex I, and so have been able to assess the effects of any changes on complex I directly, under exactly specified conditions (pH, NADH, NAD⁺, O₂ concentrations etc.) and in the absence of other cellular components. *In vivo*, the rate of H₂O₂ production is also determined by the status of the NAD⁺ pool and the local O₂ concentration, which becomes more reduced and increases, respectively, as complex I catalysis decreases (as occurs in *NDUFS4* knockout cells), acting to increase the rate. Therefore, the outcome *in vivo* is a combination of effects, not all of which were captured in our study. We have drawn attention to this fact on line 476, following our discussion on this topic:

“Increased complex I-linked ROS in mitochondria may result from changes to the environment, as well as from the complex I molecule itself, as a result of increased reduction of the NAD⁺ pool and/or increased local O₂ concentration (PMID: 16682634).”

line 425-426: it was demonstrated in the literature that MEFs derived from the *Ndufs4*^{-/-} mouse display rotenone-sensitive oxygen consumption, meaning that CI is active *in situ* (PMID: 22430089).

We thank the reviewer for bringing our attention to this result. We agree that it supports our statement that the ‘*ndufs4*^{-/-} enzyme is intact *in vivo*’ and therefore we have now added the relevant reference (line 434).

line 530/531 and 535-536: how do the authors explain the presence of NDUFAF2 in active CI on BN-PAGE in patient fibroblasts (PMID: 32335026)? This strongly argues that NDUFAF2 acts as a stabilizing bridge.

In PMID: 32335026, NDUFAF2 was found associated with the CI-830 subcomplex rather than with intact complex I in BN-PAGE analyses of NDUFS4 patient material, which is consistent with our complexomic analyses. It appears that, in the absence of NDUFS4, NDUFAF2 is not sufficiently stabilizing to keep the N-module attached during electrophoresis.

MINOR COMMENTS:

line 35-36: Does CI also contribute to ROS generation during non-pathological conditions?

ROS production by CI depends on the cellular conditions, including the potential of the NAD⁺ and Q pools, the local O₂ concentration and the proton-motive force – whenever O₂ is present it will occur, including in healthy tissues, but it increases under pathological conditions that affect respiratory chain turnover or induce reverse electron transport. However, this is not a focus of our current manuscript and we have referred to this matter only briefly, where required for evaluation of our data, for example on line 476 as noted above. In the introduction (lines 33-34) we now refer briefly to “Together with the ability of complex I to catalyze reactive oxygen species production ...”

line 42-44: Please state that the NDUFAB1 subunit occurs twice in the mature complex.

Thank you, we have now included the following text on line 42: “and 31 supernumerary subunits, including two copies of NDUFAB1..”

Reviewer 2

The manuscript by Yin and colleagues reports cryoEM structures of complex I/complex I assembly intermediates isolated from a *ndufs4* knock-out mouse line. In humans, mutations in NDUFS4 are a frequent cause of Leigh syndrome. The migration behavior of complex I and assembly factors in mitochondrial membranes was carefully analyzed by complexome profiling. The most prominent species was the QP complex lacking the N module. A complex I variant with all modules was a minor species. Assembly factor NDUFAF2 comigrated with the QP subcomplex.

It is surprising that also ACAD9 appears to be associated with this subcomplex. ACAD9 is required for beta oxidation but is also a central component of the MCIA complex, together with ECSIT and TMEM126B. The latter two subunits are mainly found in the region of supercomplexes and separated from ACAD9. A comment on this unexpected behavior would be welcome. It may also help to give a bit more background on ACAD9 and ACADVL in the manuscript. Please also note that ACAD9 is deflavinated when bound to ECSIT.

Medical Research Council Mitochondrial Biology Unit

University of Cambridge, The Keith Peters Building, Cambridge Biomedical Campus, Hills Road, Cambridge CB2 0XY

T: +44 (0)1223 252700 mrc-mbu.cam.ac.uk [@MRC_MBU](https://twitter.com/MRC_MBU)

It was indeed an unexpected result to see ACAD9 associated with the ~800 kDa band. We thank the reviewer for suggesting we evaluate this observation further and have now added the following text at lines 167-175 in the revised manuscript:

“We note that ACAD9, in its active FAD-bound form, plays an important role in beta-oxidation of fatty acids in the mitochondrial matrix, but the deflavinated form of ACAD9 also forms part of the mitochondrial complex I intermediate assembly (MCIA) complex (containing NDUFAF1, ECSIT and TMEM126B) that is expected to interact with newly translated subunit ND2 during assembly of the complex I QP module (PMID: 32320651). No intact MCIA was observed in our analyses (the other components are observed comigrating at a higher apparent mass in the *ndufs4*^{-/-} sample) and the comigration of ACAD9 with the ~800 kDa intermediate is therefore unexpected.

We have also added a brief introduction to ACADVL on lines 248-250 of the revised manuscript:

“ACADVL is also required for beta-oxidation (PMID: 18227065) but it has not previously been reported to associate with complex I, or to play a role in respiratory complex assembly.”

Electron transfer activity with different acceptors and ROS formation were determined. The authors did not detect an increased formation of ROS. However, I miss a comment here on how specific their assay is. The Amplex red test monitors H₂O₂ but would it also detect superoxide? This question seems important to me to understand the pathogenesis of NDUFS4 dysfunction and why exposure to low oxygen concentrations is beneficial. Are there alternatives for the Amplex red assay?

We understand the need to clarify this point and have added the following text at lines 237-240 of the revised manuscript to better explain the species our assay is measuring:

“The rate of H₂O₂ production from the reaction of the NADH-reduced flavin with O₂ (determined using Amplex Red to detect both the H₂O₂ produced directly and the H₂O₂ produced by spontaneous dismutation of superoxide) (PMID: 16682634) also decreased for *ndufs4*^{-/-} relative to wild-type complex I.”

The Amplex Red assay is a quantitative assay that we use routinely to measure total H₂O₂ production by complex I. Previously we have also used the reduction of acetylated cytochrome c specifically to measure superoxide, but the assay is technically more difficult and sufficient material to ensure reliable results was not available in this study.

For structure determination complex I was purified by chromatography in detergent. Cross-linking stabilized the fragile complex(es) and permitted to determine structures at up to 2.9 Å resolution. Given the small sample size, this is impressive. However, cross-linking may have also caused tight binding of the ACADVL close to the "heel" of the L shaped particle (class 1) which is discussed as a possible artefact. In line with a previous publication, NDUFA12 was not detected in any of the three classes and the authors propose that mainly NDUFS6 is responsible for detachment of the

NDUFAF2 assembly factor in the mutant (Fig. 7). It is discussed that in the NDUF54 knock-out, NDUFA12 is greatly diminished (ref Adjobo-Hermans et al.). However, according to the complexome profiling analysis in this study, this seems not to be the case (or normalization of signal strength in the heat map needs more explanation). NDUFA12 is present and migrates as a lower molecular weight species. Please comment. And it would help to somehow improve the labelling in Fig. EV1.

We thank the reviewer for bringing this apparent discrepancy to our attention. It is correct that NDUFA12 was detected in the complexome of the knockout mitochondrial membranes. However, in complexome analyses, the data for every protein is normalised to 1 for the gel-slice which has the highest abundance, and therefore the data do not reflect the absolute abundance of the protein, only the relative intensity within the lane. In our additional data on label-free quantification of the kidney mitochondrial membrane proteins, NDUFA12 was detected in both samples (7-8 peptides in the control wild-type replicates and 2-3 in the knockout replicates), but there was no single peptide common to all replicates, precluding reliable quantification. Overall, we consider our data consistent with diminished content of NDUFA12, and in this sense the complexome profiling is potentially misleading. We agree that the normalisation procedure should have been described more clearly in the text and so have added the following sections of text for better explanation of this topic:

Lines 161-163: "Note that the complexome profiles for each protein are normalised to the intensity in the highest abundance slice in each lane so they do not reflect the absolute abundance, only the relative abundance within the lane."

Lines 197-199: "Although NDUFA12 peptides were detected in both wild-type and ndufs4-/- samples, fewer peptides were detected for ndufs4-/- and no peptides were in common to all replicates, precluding quantification."

We also noted specifically the "loss of more than 90% of subunit NDUFA12" observed previously on lines 114-115, and we have improved the labelling of Figure EV1 (and adjusted the legend) as requested.

Taken together, this is a very well written manuscript that offers new insight into mitochondrial disease and complex I assembly based on excellent cryoEM data.

We thank the reviewer for their positive assessment of our work.

Reviewer 3

In this manuscript, Yin et al describe the Cryo-EM structure of the complex I species present in mouse mitochondria that lack the subunit Ndufs4. This is particularly relevant, as this genetic mouse model is one of the few animal models that is available for studying complex I deficiency (particularly as a model of Leigh Syndrome) and has been used extensively in the field of mitochondrial disease and complex I biology. It is established that in the absence of the subunit NDUF54, the catalytic N-module of complex I is destabilised from the rest of the complex by blue native PAGE analysis

Medical Research Council Mitochondrial Biology Unit

University of Cambridge, The Keith Peters Building, Cambridge Biomedical Campus, Hills Road, Cambridge CB2 0XY

T: +44 (0)1223 252700 mrc-mbu.cam.ac.uk [@MRC_MBU](https://twitter.com/MRC_MBU)

(Calvaruso et al 2012; Stroud et al 2016), yet there is still residual complex I activity detected in these mitochondria. Here, the authors address this by using chemical crosslinking on isolated mitochondria and demonstrate that in vivo, the N module remains mostly attached and that this crosslinked complex I can be used for further structural determination. The structural studies identified several classes of complex I intermediates in the absence of NDUFS4, notably including a structure with the complex I assembly factor NDUF2.

Overall, this manuscript provides structural detail for an important mitochondrial disease-causing mutation, however, this reviewer feels that many statements made are over-interpreted in the context of the data presented and the absence of further biochemical data. In order to be considered for publication in EMBO Journal, the authors should address the following points

Major points:

1. The results shown in figure EV1 should include the subunit/assembly factors as part of the figure to identify each protein being described easily.

Thank you for this helpful suggestion to improve the ease of interpretation of Figure EV1. We have now added labels on to the figure to enable the individual subunits and proteins to be more easily identified.

2. The structure of Ndufaf2 in the Heart class 3 is very interesting (Figure 5B). Given that a substantial amount of the protein density is present in the CryoEM structure, can the authors use the alphafold model for Ndufaf2 to predict the missing density?

This is an excellent suggestion, thank you. We have now added the alpha-fold predicted structure to Figure 5C for direct comparison (with an explanation of the model-confidence colour scale provided in the legend), and we have also added a comment to the results section at lines 345-350:

“Our experimental structure of NDUF2 matches well with the AlphaFold prediction (PMID: 34265844), with regions predicted but not observed in our structure (an N-terminal alpha-helix, loop between the diagonal helix and the C-terminal beta-hairpin, and a C-terminal coil) only predicted with low confidence, and with AlphaFold scores indicating they may be disordered (Figure 5C). The location and fold of NDUF2, including regions that are not well-ordered, also matches well...”

3. The discussion surrounding the presence of the ACAD proteins present in the Cryo-EM structure is quite confusing and convoluted for a result that is artefactual. The authors should consider revising this section to improve clarity to a broader audience.

In order to improve the clarity and accessibility of our presentation on the ACAD proteins we have provided additional background information on both ACAD9 (lines 167-175) and ACADVL (lines 248-250) as follows:

“We note that ACAD9, in its active FAD-bound form, plays an important role in beta-oxidation of fatty acids in the mitochondrial matrix, but the deflavinated form of ACAD9 also forms part of the mitochondrial complex I intermediate assembly (MCIA) complex (containing NDUFAF1, ECSIT and TMEM126B) that is expected to interact with newly translated subunit ND2 during assembly of the complex I QP module (PMID: 32320651). No intact MCIA was observed in our analyses (the other components are observed comigrating at a higher apparent mass in the ndufs4-/- sample) and the comigration of ACAD9 with the ~800 kDa intermediate is therefore unexpected.

“ACADVL is also required for beta-oxidation (PMID: 18227065) but it has not previously been reported to associate with complex I, or to play a role in respiratory complex assembly.”

We have also added an additional sentence at lines 388-391 “Furthermore, the assembly factor ACAD9, as part of the MCIA complex, is expected to be deflavinated and to interact with subunit ND2 (PMID: 32320651), which is inconsistent with our flavinated ACADVL interacting with subunit ND1” and we have carefully re-read this section of our manuscript and removed unnecessary details to decrease the length and improve the overall clarity.

4. In the discussion, the authors state that the binding of NDUFS4, NDUSF6 and NDUFA12 is required to displace NDUFAF2 (line 540). In two studies using NDUFA12 patient-derived cells or modified human cells lacking NDUFA12, it appears that complex I can assemble, is active and has similar migration to wildtype complex I and stably bind to NDUFAF2 (Adjobo-Hermans et al 2020; Stroud et al 2016). To me, this would suggest that NDUFAF2 displacement predominantly requires NDUFA12 recruitment, as NDUFAF2 can still bind to complex I in the presence of NDUFS4 and NDUSF6. The authors should include this in the context of their structural data.

We thank the reviewer for bringing this to our attention. Although the NDUFA12 studies show that NDUFA12 is important for displacing NDUFAF2, in our structures we observe that NDUFS4, NDUSF6 and NDUFA12 all overlap structurally with NDUFAF2, indicating that NDUFS4 and NDUSF6 may also contribute to destabilising it. In comparison, the contribution from NDUFS4 may be relatively small, however, NDUSF6 was implicated previously in the displacement step and it has been suggested that NDUFA12 and NDUSF6 may bind together (although our current data do not substantiate this idea). Here, we show NDUFAF2 and NDUSF6 are both present in our preparation, but our structures show they are bound to separate subpopulations and that NDUSF6 is incompletely associated. Similarly, a population of enzyme containing NDUFAF2, NDUFS4 and NDUSF6 may also comprise a mixture of subpopulations. We have now expanded our comment on lines 524-526 as follows to specifically note these NDUFAF2 studies:

“Comigration of NDUFAF2 with complex I in BN-PAGE analyses of NDUFA12-knockout cells (PMID: 32335026; PMID: 27626371) has indicated NDUFA12 is essential for the displacement step.”

5. The role of NDUFAF2 is elusive in complex I assembly, though patients with homozygous null mutations in this protein can still assemble complex I (to a lesser extent than control cells; Ogilvie et al 2005). While NDUFAF2 might participate in N-module recruitment, it cannot be excluded that

other proteins also participate in this process, given the clinical and biochemical evidence available. The authors should include this in their discussion.

Thank you for bringing this paper to our attention. We now note on lines 545-547 of the revised manuscript:

“Our proposal explains how NDUFAF2 increases the efficiency of (but is not essential for (PMID: 16200211)) N-module attachment and complex maturation...”

Minor points:

1. It is unclear where the structures in Figure 1 come from- please cite the appropriate publication and PDB ID in text/figure legend.

We thank the reviewer for bringing this oversight to our attention and have now included the PDB code for the structure used (6ZR2) in the legend for Figure 1.

2. Line 153-154: Four proteins were not detected by mass spectrometry, however, only three subunits were mentioned. What was the fourth subunit?

Only subunits ND3, ND4L and ND6 were not detected in either wild-type or knock-out complexome samples (as listed in the figure legend) and overall, 41 unique protein subunits were detected, but these account for 42 subunits of the 45-subunit complex as there are two copies of NDUFAF1. This information has been added to the introduction in line 42 of the revised manuscript:

“.. 31 supernumerary subunits, including two copies of NDUFAF1..”

Dear Dr Hirst,

Thank you for submitting a revised version of your manuscript. Your study has now been seen by all original referees, who find that their previous concerns have been addressed and now recommend publication of the manuscript.

There are two minor points raised by referee #1 and referee #3:

Referee #1

"The authors have properly addressed most of my comments.

I fully agree with the rebuttal regarding the Ndufs4 patient cells. However, this was not the point I wanted to make. Using an Ndufa12 patient cell line (PMID 323335026) evidence was provided (Fig 3G; arrowhead) that in the absence of Ndufa12 (Fig. 3E) and presence of Ndufs4 (Fig. 3D), Ndufaf2 is attached to a catalytically active complex I. The relevance and impact of this finding for the current study should be discussed. This directly relates to comment 4 raised by reviewer 3.

Yin et al have provided a revised manuscript with improved clarity for a broader audience. The manuscript revisions have addressed all my major concerns."

Referee #3

"With regard to my second minor point below and the author's response:

2. Line 153-154: Four proteins were not detected by mass spectrometry, however, only three subunits were mentioned. What was the fourth subunit?

Only subunits ND3, ND4L and ND6 were not detected in either wild-type or knock-out complexome samples (as listed in the figure legend) and overall, 41 unique protein subunits were detected, but these account for 42 subunits of the 45-subunit complex as there are two copies of NDUFAB1. This information has been added to the introduction in line 42 of the revised manuscript:

".. 31 supernumerary subunits, including two copies of NDUFAB1.."

It is my understanding that lines 152-154 should read 'For the wild-type membranes, 41 (not 40 as mentioned in the manuscript) of the 44 complex I proteins were detected in the ~1 MDa complex I band. Only subunits ND3, ND4L and ND6, which are hydrophobic proteins with few tryptic peptides, were not detected.

I congratulate the authors on this important work to understand the structural basis of disease in the case of NDUFS4 mutations."

In addition, there remain a few mainly editorial points that have to be addressed before I can extend formal acceptance of the manuscript:

1. CRediT has replaced the traditional author contributions section because it offers a systematic, machine-readable author contributions format that allows for more effective research assessment. Please remove the Authors Contributions from the manuscript and use the free text boxes beneath each contributing author's name in our online submission system to add specific details on the author's contribution. More information is available in our guide to authors.
2. Please check that the funding information is also entered into our online system and identical to the manuscript.
3. Please submit up to five keywords.
4. Please call out Appendix Figures and Tables out correctly as Appendix Figure S1-S5 and Appendix Table S1-S4
5. Please reorganize source data files to one file/folder per figure and ZIPing for each main figure. For EV and/or appendix figures, ZIP together all source data.
6. Please upload the synopsis text and image (with the correct size (550 pixels wide and 300-600 pixels high)) individually.
7. Please ad a separate 'Data Information' section is required in the legends of figures EV2.
8. Please change figure legends from 3a, b to 3A-B.
9. Please label the legends of figure EV3; EV4 (labels A, B are not provided in the legend).
10. Please rename the movie file to Movie EV1 with the corresponding callout, and the legend should be removed from the ms file, and zipped with the movie file.
11. Please ad subject categories.

With best regards,

Cornelius Schneider

Cornelius Schneider, PhD
Editor
The EMBO Journal
c.schneider@embojournal.org

Dear Editor,

Many thanks for your recent emails on our paper – we have addressed the few remaining points in our manuscript as detailed below (changes in the manuscript file are in red), and very much hope that you will now be able to proceed to the formal acceptance of our work for publication.

Best wishes,

Judy Hirst (on behalf of all authors)

Referee #1

"The authors have properly addressed most of my comments. I fully agree with the rebuttal regarding the Ndufs4 patient cells. However, this was not the point I wanted to make. Using an Ndufa12 patient cell line (PMID 323335026) evidence was provided (Fig 3G; arrowhead) that in the absence of Ndufa12 (Fig. 3E) and presence of Ndufs4 (Fig. 3D), Ndufaf2 is attached to a catalytically active complex I. The relevance and impact of this finding for the current study should be discussed. This directly relates to comment 4 raised by reviewer 3.

We appreciate the reviewer's viewpoint that the presence of a catalytically active complex I in NDUFA12 patient lines is relevant to our work and have amended our text as follows (lines 528-535):

"Comigration of NDUFAF2 with complex I in BN-PAGE analyses of NDUFA12-patient/knockout cells (PMID: 32335026; PMID: 27626371) has indicated NDUFA12 is essential for the displacement step, but not for N-module attachment. Although the overall catalytically-competent complex I population present in the patient-derived cell line contains NDUFS4, NDUFAF2 and the N-module (PMID: 32335026) but, in light of our structural data, it is not known whether they are all present in a single homogenous population, or whether different protein combinations are present in different subpopulations."

We have also added a further statement on lines 562-3 to underline the point that: "The most substantial overlap is with NDUFA12, consistent with its importance in the displacement step (PMID: 32335026; PMID: 27626371)".

Referee #3

"With regard to my second minor point below and the author's response:

2. Line 153-154: Four proteins were not detected by mass spectrometry, however, only three subunits were mentioned. What was the fourth subunit?

Only subunits ND3, ND4L and ND6 were not detected in either wild-type or knock-out complexome samples (as listed in the figure legend) and overall, 41 unique protein subunits were detected, but these account for 42 subunits of the 45-subunit complex as there are two copies of NDUFAB1. This information has been added to the introduction in line 42 of the revised manuscript:

".. 31 supernumerary subunits, including two copies of NDUFAB1.."

It is my understanding that lines 152-154 should read 'For the wild-type membranes, 41 (not 40 as mentioned in the manuscript) of the 44 complex I proteins were detected in the ~1 MDa complex I band. Only subunits ND3, ND4L and ND6, which are hydrophobic proteins with few tryptic peptides, were not detected.

The reviewer is correct. We have corrected the mistake accordingly (changed 40 to 41).

Editorial points

1. CRediT has replaced the traditional author contributions section because it offers a systematic, machine-readable author contributions format that allows for more effective research assessment. Please remove the Authors Contributions from the manuscript and use the free text boxes beneath each contributing author's name in our online submission system to add specific details on the author's contribution. More information is available in our guide to authors.

Done

2. Please check that the funding information is also entered into our online system and identical to the manuscript.

Done

3. Please submit up to five keywords.

Keywords have been added to the manuscript title page: Complex I, Cryo-EM, Leigh Syndrome, Mitochondria, NADH:ubiquinone oxidoreductase.

4. Please call out Appendix Figures and Tables out correctly as Appendix Figure S1-S5 and Appendix Table S1-S4

Done

5. Please reorganize source data files to one file/folder per figure and ZIPing for each main figure. For EV and/or appendix figures, ZIP together all source data.

Done

6. Please upload the synopsis text and image (with the correct size (550 pixels wide and 300-600 pixels high)) individually.

Done

7. Please ad a separate 'Data Information' section is required in the legends of figures EV2.

Question raised, editor confirmed no change required.

8. Please change figure legends from 3a, b to 3A-B.

Question raised, editor confirmed no change required.

9. Please label the legends of figure EV3; EV4 (labels A, B are not provided in the legend).

Done

10. Please rename the movie file to Movie EV1 with the corresponding callout, and the legend should be removed from the ms file, and zipped with the movie file.

Done

11. Please ad subject categories.

Question raised, editor confirmed no additions required.

Dear Prof. Hirst,

I am pleased to inform you that your manuscript has been accepted for publication in the EMBO Journal.

Yours sincerely,

Cornelius Schneider, PhD
Editor
The EMBO Journal
c.schneider@embojournal.org
